# Health-Related Quality of Life and Side Effects in Gastrointestinal Stromal Tumor (GIST) Patients Treated with Tyrosine Kinase Inhibitors: A Systematic Review of the Literature

**DOI:** 10.3390/cancers14071832

**Published:** 2022-04-05

**Authors:** Deborah van de Wal, Mai Elie, Axel Le Cesne, Elena Fumagalli, Dide den Hollander, Robin L. Jones, Gloria Marquina, Neeltje Steeghs, Winette T. A. van der Graaf, Olga Husson

**Affiliations:** 1Department of Medical Oncology, The Netherlands Cancer Institute-Antoni van Leeuwenhoek, 1066 CX Amsterdam, The Netherlands; d.vd.wal@nki.nl (D.v.d.W.); n.steeghs@nki.nl (N.S.); w.vd.graaf@nki.nl (W.T.A.v.d.G.); 2Department of Medical Oncology, Radboud University Medical Center, 6525 GA Nijmegen, The Netherlands; mai.elie@ru.nl (M.E.); d.denHollander@radboudumc.nl (D.d.H.); 3Department of Medical Oncology, Gustave Roussy, 94805 Villejuif, France; axel.lecesne@gustaveroussy.fr; 4Department of Medical Oncology, IRCCS Foundation National Cancer Institute, 20133 Milan, Italy; elenaRosa.Fumagalli@istitutotumori.mi.it; 5Department of Clinical Oncology, The Royal Marsden Hospital and Institute of Cancer Research, London SM2 5 NG, UK; robin.jones4@nhs.net; 6Department of Medical Oncology, Hospital Clinico San Carlos, 28040 Madrid, Spain; gloria.marquina@salud.madrid.org; 7Department of Clinical Pharmacology, The Netherlands Cancer Institute-Antoni van Leeuwenhoek, 1066 CX Amsterdam, The Netherlands; 8Department of Medical Oncology, Erasmus MC Cancer Institute, Erasmus University Medical Center, 3015 GD Rotterdam, The Netherlands; 9Department of Psychosocial Research and Epidemiology, The Netherlands Cancer Institute, 1066 CX Amsterdam, The Netherlands; 10Department of Surgical Oncology, Erasmus MC Cancer Institute, Erasmus University Medical Center, 3015 GD Rotterdam, The Netherlands; 11Division of Clinical Studies, Institute of Cancer Research, London SM2 5NG, UK

**Keywords:** gastrointestinal stromal tumor, tyrosine kinase inhibitor, health-related quality of life, adverse events, side effects, patient-reported outcome measures

## Abstract

**Simple Summary:**

Gastrointestinal stromal tumors (GISTs) are the most common mesenchymal tumors of the gastrointestinal tract, mostly driven by activating mutations in *KIT* or *PDGFRα* oncogenes. The introduction of tyrosine kinase inhibitors (TKIs) has revolutionized the treatment of GIST resulting in a substantial gain in median overall survival. Nowadays, imatinib, sunitinib, regorafenib, and ripretinib are registered as first, second, third, and fourth-line therapies, and avapritinib is registered specifically for GISTs harboring a *PDGFRα* exon 18/D842V mutation. As a result, health-related quality of life (HRQoL) has become more relevant for this surviving population, which is increasing in number. In daily clinical practice, the side effects of TKIs and their impact on the daily lives of patients are the main reason for treatment adjustments. Currently, an overview of HRQoL issues and side effects of different TKIs registered for the treatment of GIST is lacking.

**Abstract:**

Background: The introduction of tyrosine kinase inhibitors (TKIs) has revolutionized the treatment of gastrointestinal stromal tumors (GISTs), resulting in a substantial gain in median overall survival. Subsequently, health-related quality of life (HRQoL) has become more relevant. Here, we systematically review the available literature on HRQoL issues and side effects of different TKIs registered for the treatment of GIST. Methods: A search through five databases was performed. Full reports in English describing HRQoL outcomes and/or side effects in GIST patients on TKI therapy were included. Results: A total of 104 papers were included; 13 studies addressed HRQoL, and 96 studies investigated adverse events. HRQoL in patients treated with imatinib, regorafenib, and ripretinib remained stable, whereas most sunitinib-treated patients reported a decrease in HRQoL. Severe fatigue and fear of recurrence or progression were specifically assessed as HRQoL issues and had a negative impact on overall HRQoL as well as psychological and physical well-being. The majority of studies focused on physician-reported side effects. Nearly all GIST patients treated with a TKI experienced at least one adverse event, mostly mild to moderate. Conclusions: Despite the fact that almost all patients treated with a TKI experienced side effects, this did not seem to affect overall HRQoL during TKI therapy. In daily practice, it are the side effects that hamper a patient’s HRQoL resulting in treatment adjustments, suggesting that the reported side effects were underestimated by physicians, or the measures used to assess HRQoL do not capture all relevant issues that determine a GIST patient’s HRQoL.

## 1. Introduction

Gastrointestinal stromal tumors (GISTs) are the most common mesenchymal tumors of the gastrointestinal tract, affecting 10–20 people per million per year [1]. The diagnosis of GIST relies on morphology and positive immunohistochemistry for CD117 (KIT) and/or DOG1 [2,3]. Most GISTs are driven by activating mutations in *KIT* (75%) or *PDGFRα* (15%) oncogenes [4,5,6]. The mainstay of treatment for localized GISTs is complete surgical resection. However, it is not uncommon for GIST to have already metastasized to the peritoneum or liver at time of diagnosis. Before the introduction of imatinib, the median survival of patients with metastatic GIST was only a year. In 2001, imatinib was granted accelerated approval by the United States Food and Drug Administration (FDA) as first-line therapy for advanced and metastatic GIST based on response rate [7]. Imatinib significantly changed the prognosis of metastatic GIST patients to a median survival of 57 months [8]. Today, 10–15% of imatinib-treated patients with metastatic GIST are still responding to imatinib after 10 years of treatment [9]. Nevertheless, in most metastatic GIST patients, the disease progresses after 24 months [8,9], often caused by secondary mutations in the *KIT* gene [10,11]. In 2007, sunitinib was approved as second-line therapy for GIST based on improvement in time to progression shown in an interim efficacy analysis [12]. The FDA approved regorafenib as third-line therapy for GIST in 2013 based on significantly improved progression-free survival [13]. Recently, ripretinib was registered as fourth-line therapy [14], and avapritinib was approved, specifically for GISTs harboring the *PDGFRα* exon 18/D842V mutation [15,16]. After the success of imatinib in the treatment of advanced and metastatic GIST, imatinib was also approved as an adjuvant treatment for 3 years in patients with high-risk disease in 2008 [17,18].

As a result of the extended survival, aspects regarding health-related quality of life (HRQoL) have become more relevant. HRQoL is a multidimensional concept that includes the patient’s perception of the impact of the disease and its treatment on physical, psychological, and social functioning [19]. Clinical outcomes (e.g., response rate, time to progression, progression-free survival) together with patient-reported outcomes are needed to determine the net clinical benefit of TKIs. Nevertheless, until recently, FDA approvals, also for imatinib in both metastatic and adjuvant setting, sunitinib and regorafenib, were only based on objective or physician-reported data.

Overall, TKIs, especially imatinib, are described as tolerable with manageable side effects [20]. Most studies present physician-reported adverse events (AEs), while tools are available to collect patient-reported side effects (e.g., patient-reported outcomes version of the Common Terminology Criteria for Adverse Events (PRO-CTCAE) or MD Anderson Symptom Inventory). In addition to the physical side effects of TKIs, psychological and social challenges may also influence the daily lives of GIST patients. As most patients with metastatic GIST eventually succumb to their disease [9,21], the fear of disease progression is undeniably a challenge for most patients [22]. Metastatic GIST patients may struggle with the side effects of TKI therapy and the consequences of living with cancer with an extended survival time [23]. Currently, an overview of patient-reported HRQoL issues and AEs for GIST patients on TKI therapy is lacking. Here, we systematically review the available literature on HRQoL issues and side effects of different TKIs registered for the treatment of GIST.

## 2. Materials and Methods

The protocol for this systematic review was registered on PROSPERO (CRD42021227658) and followed the Preferred Reporting Items for Systematic Reviews and Meta-Analyses (PRISMA) guidelines [24].

### 2.1. Search Strategy

A systematic literature search through Medline, Embase, PsycINFO, Web of Science, and Cochrane Library was performed on 21 December 2020. The search string combined terms for GIST and TKI, including specific TKI therapies, and HRQoL or side effects. Terms for qualitative research were added to ensure that all research regarding HRQoL was found. The full search string is presented in Appendix A. There was no restriction regarding year of publication, as all found literature was published after 2001 when imatinib was approved by the FDA.

### 2.2. Selection Process

Studies were eligible for inclusion if; (1) they included GIST patients on TKI therapy in neo-adjuvant, adjuvant, or advanced/metastatic setting; (2) the objective was to describe HRQoL outcomes, side effects, adverse events; (3) they were full reports published in English. Studies were excluded if; (1) they were phase 1 studies, individual case reports, small case series (<10 patients), conference proceedings, or abstracts, (2) the study sample consisted of multiple cancer types, including GIST, but the data for patients with GIST could not be extracted, (3) patients were treated with non-registered TKIs or data about registered and non-registered TKIs were not presented separately. Two independent reviewers (D.v.d.W., M.J.P.R.) screened all hits on title and abstract for eligibility. Papers selected by both reviewers were included, and papers selected by one of the reviewers were discussed for consensus. One reviewer (D.v.d.W.) read the full text and extracted all relevant data from the included papers. Reviews and meta-analyses were used for cross-referencing purposes only and were excluded afterward for not reporting ‘primary data’.

### 2.3. Quality Assessment

The methodological quality of the selected papers was independently assessed by two reviewers (D.v.d.W., M.J.P.R) using the Mixed Method Appraisal Tool (MMAT) [25]. This tool was considered the best fit for this review as it is designed to appraise a variety of study designs. For each included study a total score was calculated ranging from 1 * to 5 *; 1 * indicating a study of poor quality and 5 * indicating a study of good quality. A detailed quality assessment of the included studies is presented in Appendix A. 

### 2.4. Data Extraction

Study design, aim, patient characteristics, type of TKI, treatment duration, used HRQoL or side effects measure(s), and results were extracted from the included papers. We categorized studies regarding HRQoL by type of study design, including longitudinal, cross-sectional, and qualitative designs. Studies about side effects were stratified per type of TKI into the categories: imatinib, sunitinib, regorafenib, and ripretinib. If a paper reported on different types of TKIs, the results were included for each category separately. In case of imatinib therapy, papers were also sorted by treatment setting.

## 3. Results

### 3.1. Sample

The literature search yielded 3455 unique hits; after screening on title and abstract, 264 papers met our criteria for full-text review. After an independent full-text review, 104 papers were included. The flow chart of the selection procedure is shown in Figure 1. 

### 3.2. Study Characteristics

All 104 papers were published between 2002 and December 2020. Of the included studies, 8 studies addressed HRQoL, 5 studies reported on both HRQoL and adverse events, and 91 studies focused on adverse events only.

### 3.3. Health-Related Quality of Life (HRQoL)

Thirteen studies assessed HRQoL or a specific HRQoL issue. Studies had a prospective (*n* = 8), retrospective (*n* = 3) or qualitative (*n* = 2) design. The study characteristics and main findings are summarised in Table 1. Fifteen different patient-reported outcome measures (PROMs) were used, of which the EORTC QLQ-C30 (*n* = 9) was the most commonly used questionnaire. The characteristics of the used measures are described in Table 2.

#### 3.3.1. Imatinib

Four prospective studies reported on HRQoL during imatinib therapy; one study addressed GIST patients in the adjuvant setting, and three studies GIST patients in the palliative setting. In 40 GIST patients receiving adjuvant imatinib, QoL remained stable throughout 60 months of treatment [26]. A cross-sectional study [27] evaluated imatinib adherence in 158 adjuvant GIST patients, 58% of the patients were considered nonadherent, and nonadherence was associated with a low global QoL score. A multicenter study [28] compared global health status in unresectable and metastatic GIST patients at baseline and after 12 months of imatinib therapy. Global health status did not vary significantly, with 20, 15, and 16 of the 51 patients experiencing an improvement, stable, or worsening in global health status, respectively. Subsequently, patients with controlled disease after 1 year of imatinib interrupted or continued imatinib treatment. At 6 months of follow-up, no differences were observed in global health status, functional status, or symptoms scales between the interruption and continuation group [28]. An observational study [29] in a real-world setting reported that after 18 months of imatinib, overall HRQoL was stable in 51.5% and had improved in 25.8% of the 77 GIST patients with unresectable or metastatic disease. In a prospective single-center study [30], metastatic patients received imatinib or placebo after pre-treatment with at least imatinib and sunitinib. At 8 weeks of treatment, there were no differences in global QoL and functioning scales. Cross-sectionally, the pain was significantly better, while nausea, vomiting, appetite loss, and diarrhea were worse in the imatinib group.

#### 3.3.2. Sunitinib

One prospective study [31] assessed HRQoL in sunitinib-treated patients. In this study, 44 patients were treated with either sunitinib or masitinib after progression on imatinib. Global QoL was stable or improved in 5 of the 13 sunitinib-treated patients, but the timing of the longitudinal HRQoL assessment was unclear.

#### 3.3.3. Regorafenib

In a multicenter randomized controlled trial [32], 122 patients received regorafenib. Health utility scores remained stable; neither cycle number nor treatment type (off-treatment vs. regorafenib; placebo vs. regorafenib) significantly influenced health utility. However, confirmed disease progression led to a significantly impaired health status.

#### 3.3.4. Ripretinib

One international randomized controlled trial [33] reported on HRQoL in 129 GIST patients receiving ripretinib or a matching placebo. Overall health, role, and physical functioning remained stable in the ripretinib group compared with a decrease in the placebo group from baseline to cycle two, day 1.

**Table 1 cancers-14-01832-t001:** Studies reporting on HRQoL.

Author, Year, Country (Ref)	Design	Aim	Patients Characteristics (Number of Patients, Gender Male %, Age Range)	Treatment/Intervention	Outcome Measure	Results	Quality Score
Raut, 2018, USA [26]	Prospective, multicenter, phase II	To determine whether adjuvant treatment for primary GIST with imatinib for 5 years is tolerable and efficacious	91 patients with intermediate or high risk of recurrence after resection of primary GIST, 53% male, median age of 60 years (range 30–90)	Imatinib 400 mg/day for a median duration of 55.1 months (range 0.5–60.6), 46 patients completed the intended 5 years of treatment	FACT-G	In 40 patients who continued follow-up and completed surveys, QoL remained stable and never decreased more than 3 points from baseline throughout 60 months of imatinib treatment. Surveys were not completed by patients after early discontinuation of treatment.	3 *
Blay, 2007, France [28]	Prospective, multicenter, randomized, phase III	To determine whether interruption of imatinib is feasible in advanced GIST patients with controlled disease after 1 year of imatinib	182 patients with advanced GIST, 59% male, median age of 62 years (range 27–87)	Imatinib 400 mg/day for 12 months	QLQ-C30	98 patients had controlled disease after 1 of imatinib therapy, and the QLQ-C30 was returned by 56 of the 98 patients both at baseline and at month 12. Global health status did not vary significantly with 20, 15, and 16 patients experiencing an improvement, stable or worsening in global health status.	2 *
			58 patients with unresectable or metastatic GIST, 62% male, median age of 61 years (range 27–83)	32 patients interrupted and 26 patients continued imatinib 400 mg/d after 1 year	QLQ-C30	At 6 months of follow-up, 13 of the 32 patients in the interruption group and 16 of the 26 patients in the continuation group completed the QLQ-C30. No differences were observed in global health status, functional status or symptoms scales between the two groups.	2 *
Bouche, 2018, France [29]	Prospective, multicenter, observational	To describe the profile of treated patients, the prescription patterns and the impact of treatment on population health in a real-world setting	151 patients with unresectable or metastatic GIST, 58% male, median age 60 years (range 21–86)	Imatinib 200–800 mg/day, for a median duration of 42.6 months (range 4.9–86.7)	QLQ-C30, SF-36	HRQoL data were available for 110 patients at baseline and 77 patients after 18 months of follow-up. After 18 months, 51.5% of the patients had a stable and 25.8% an improved global QoL score (QLQ-C30), with slight improvement in some mean physical and mental scores (SF-36).	3 *
Yoo, 2016, South Korea [30]	Prospective, single-center, randomized, phase III	To assess the impact of imatinib rechallenge on HRQoL in heavily pre-treated patients	81 patients with unresectable or metastatic GIST who had progressed on at least imatinib and sunitinib, 68% male, median age 59 years (range 52–67)	37 patients received imatinib 400 mg/day and 35 patients received placebo	QLQ-C30	HRQoL data were collected only during the double-blind treatment period. At 8 weeks of treatment 25 patients in the imatinib arm and 21 patients in the placebo arm were evaluable for HRQoL analysis. At 8 weeks, there were no differences in global health status/QoL and functioning scales. Cross-sectionally, pain was significantly better while nausea/vomiting, appetite loss and diarrhea were worse in the imatinib group.	4 *
Adenis, 2014, France [31]	Prospective, multicenter, randomized, phase II	To evaluate the safety and efficacy of masitinib versus sunitinib	44 advanced GIST patients with progressive disease on imatinib ≥ 400 mg/day, 50% male, mean age 64 years (range 31–85)	23 patients received masitinib 12 mg/kg/day in two daily intakes for a median of 4.7 months and 21 patients received sunitinib 50 mg/day in a 4-weeks-on-2-weeks-off schedule for a median of 3.8 months	QLQ-C30	At baseline, global QoL was good (60–65) and similar in both treatment arms. The time of longitudinal HRQoL assessment was unclear. Improved or stable global QoL was reported in 10 of the 15 patients treated with masitinib versus 5 of the 13 patients treated with sunitinib.	3 *
Poole, 2015, UK [32]	Prospective, international, multicenter, randomized, phase III	To characterize the health state utility of advanced GIST patients treated with regorafenib and to estimate the health states of patients remaining progression-free and those with clinically diagnosed progression	182 (91% of the intention-to-treat population) with advanced GIST refractory to imatinib and sunitinib, 64% male, average age 58 years	122 patients received regorafenib 160 mg daily for the first 3 weeks of every 4-week cycle and 60 patients received matching placebo for an unknown duration	EQ-5D index score	Health utility scores remained stable, neither cycle number nor treatment type (off-treatment vs. regorafenib; placebo vs. regorafenib) significantly influenced health utility. Confirmed disease progression led to a significantly impaired QoL.	5 *
Blay, 2020, France [33]	Prospective, international, multicenter, randomized, phase III	To evaluate the safety and efficacy of ripretinib as fourth-line therapy	129 advanced GIST patients with progression on at least imatinib, sunitinib, and regorafenib or intolerance to any of these therapies, 57% male, median age 61 years (range 29–83)	85 patients received ripretinib 150 mg/day in 28-day cycles for an unknown duration and 44 patients received matching placebo for an unknown duration	QLQ-C30 (only physical and role functioning questions), EQ-VAS	HRQoL data were available for 74 and 42 patients in the ripretinib and placebo group, respectively. Overall health, role and physical functioning from baseline to cycle 2 day 1 remained stable in the ripretinib group compared with a decrease in the placebo group.	5 *
Carbajal-Lopez, 2020, Mexico [34]	Prospective, multicenter, randomized	To identify and compare the effects of two online interventions in terms of fatigue, distress and quality of life	27 patients (response rate 27%) with GIST of which 21 received imatinib, 41% male, mean age 49 years	13 patients were assigned to an internet-delivered cognitive behavior therapy and 14 patients to an online psychoeducation program	QLQ-C30, MFI, HADS	Both interventions led to a reduction in fatigue and distress, and an increase in global QoL and all functioning domains of HRQoL.	3 *
Custers, 2015, The Netherlands [22]	Retrospective, cross-sectional, single center	To assess HRQoL, distress, and fear of cancer recurrence or progression (FCR)	54 (response rate 64%) patients with localized or metastatic GIST, 54% male, median age 63 years (range 21–84)	33 patients were receiving imatinib for an unknown duration	CWS, FCRI, QLQ-C30, HADS, IES	52% reported high levels of FCR, these patients had lower scores on global QoL and the subscales role, emotional, cognitive, and social functioning, experienced higher levels of psychological distress and difficulty making plans for the future.	4 *
Poort, 2016, The Netherlands [35]	Retrospective, cross-sectional, single center	To determine the prevalence of severe fatigue, impact of severe fatigue on QoL, psychosocial variables and physical functioning, and explore associations between fatigue and current TKI use	89 (response rate 75%) GIST patients, 58% male, median age 64 years (range 21–86) were compared with 234 matched healthy controls (MHC), 64% male, median age 64 years (range 18–90)	61 patients were receiving TKI treatment (imatinib (*n* = 52), sunitinib (*n* = 7), nilotinib (*n* = 2)) for an unknown duration	CIS-fatigue, QLQ-C30, HADS, SF-36, SES, FCS	Severe fatigue occurred in 30% of the GIST patients compared to 15% in the MHC. Current TKI use was associated with fatigue severity. Severe fatigued patients had a lower global QoL, increased impairment on all the functional domains, less favorable physical functioning, lower self-efficacy and more fatigue catastrophizing and psychological distress.	5 *
Wang, 2020, China [27]	Retrospective, cross-sectional, single center	To evaluate the prevalence of imatinib adherence and its influencing factors	158 GIST patients in the adjuvant setting, 56% male, median age 56 years	Imatinib 400 mg/day for a median of 11 months (range 1–152)	MMAS, QLQ-C30, SSRS	92 (58%) patients were considered nonadherent. Female gender, living in a rural area and having a low global QoL score were associated with nonadherence.	5 *
Macdonald, 2012, USA [36]	Qualitative, international, multicenter	To explore the experiences and emotions of patients through GIST diagnosis, treatment initiation, disease control, and in some patients, loss of response and therapy switch	50 patients with GIST from Canada (*n* = 15), the United States (*n* = 10), Brazil (*n* = 5), France (*n* = 5), Germany (*n* = 5), Russia (*n* = 5) and Spain (*n* = 5)	25 patients were treated in an adjuvant setting and 25 patients in a metastatic setting, type of treatment and duration were unknown	Interviews	Patients shared common experiences during each stage of disease management, namely crisis, hope, adaptation, ‘new normal’ and uncertainty. Most patients were highly self-directed and understood the importance of adherence to TKI therapy, while acknowledging that the therapy may had an impact on their daily lives.	4 *
Fauske, 2019, Norway [23]	Qualitative, single center	To explore how patients with metastatic GIST experience both living with their disease and the side effects of its treatment	20 patients with metastatic GIST, 45% male, median age 61 years (range 36–85)	18 patients were treated with imatinib 200 mg/day (*n* = 4), 400 mg/day (*n* = 12) or 800 mg/day (*n* = 2) and 2 patients with sunitinib, patients were receiving systemic treatment for a median of 6 years (range 2–15)	Interviews	More than half of the participants experienced side effects that influenced their daily lives in negative and challenging ways, which urged them to adapt to ‘a new normal’. The majority of participants reported the well-known side effects of imatinib, such as (peri-orbital) edema, nausea, diarrhea, muscle cramps, muscle aches, joint pain, tiredness and exhaustion. Many also reported an increased need for sleep, cognitive challenges, reduced sexual desire, as well as poor stress tolerance around the intake of their GIST medication. Although participants struggled with the side effects and the consequences of living with a chronic cancer, half of them considered themselves to be healthy and able to live a normal life.	5 *

* Represents the overall methodological quality of the study ranging from 1 * to 5 *; 1 * indicating a study of poor quality and 5 * indicating a study of good quality.

**Table 2 cancers-14-01832-t002:** Characteristics of the frequently used measures.

Questionnaire, Year, Reference	Full Name	General Description of the Measure	Number of Items, Score Range, Score Interpretation	Number of Studies in This Review That Have Used the Questionnaire
FACT-G, 1993 [37]	Functional Assessment of Cancer Therapy—General	Measure four domains of health-related quality of life in cancer patients: physical, social, emotional, and functional well-being.	27 items, subscale scores range from 0–28 for physical, functional and social well-being, 0–24 for emotional well-being. Add subscale scores to derive total FACT-G score, range 0–108. The higher the score, the better the QOL.	1
EORTC QLQ-C30, 1993 [38]	European Organisation for Research and Treatment of Cancer Quality of Life Questionnaire-30	Assess health-related quality of life of cancer patients. Consists of five functional scales (physical, role, emotional, cognitive and social functioning), three symptom scales, a global health status/QoL scale, and six single items.	30 items, all scales and single-items range in score from 0 to 100. A high score for a functional or global health status/QoL scale represents a high level of functioning or a high QoL. A high score for a symptom scale or item represents a high level of symptomatology or problems. Studies included in this review defined an improvement or worsening as a change of ≥10 points from baseline.	9
SF-36, 1992 [39,40]	Short-Form 36-Item Health Survey	Assesses health status on eight domains: physical functioning, role limitations due to physical health, role limitations due to emotional problems, vitality (energy and fatigue), emotional well-being, social functioning, pain, general health. Physical component score composed of four scales; physical function, role limitations due to physical health, bodily pain and general health. Mental component score composed of four scales; vitality, social functioning, role limitations caused by emotional problems and mental health.	36 items, each domain is scored on a 0 to 100 range, a high score defines a more favorable health state.	2
EQ-5D-3L, 1990 [41,42]	N/A	Measure general health status in two parts: descriptive and the EuroQol Visual Analogue Scale (EQ-VAS).		
		EQ-5D measures patient health utility (health status/QoL) using a descriptive system that assesses five generic dimensions of health: mobility, self-care, usual activity, pain and discomfort, and anxiety and/or depression.	5 items, score ranges from 1–3 for each item, these health states can be converted to a single summary score, the EQ-5D index score. According to the EQ-5D index, 1.0 represents perfect health and 0.0 represents death.	1
		The EQ-VAS records the patient’s self-rated overall health on a vertical visual analogue scale.	1 item, range 0–100 from “Worst Possible” to “Best Possible” health, higher scores represent better health.	1
HADS, 1983 [43,44]	Hospital Anxiety and Depression Scale	To assess psychological distress and detect states of anxiety and depression.	14 items divided into 2 subscales; anxiety and depression. Score ranges from 0–3 for each item. Scores for each subscale range from 0–21, a score of 11 or higher indicates a mental disorder. Higher scores indicate more anxiety, depression, and psychological distress.	2
Mexican adaptation of the HADS, 2015 [45]	Mexican adaptation of the Hospital Anxiety and Depression Scale	To evaluate distress.	12 items with 6 items on anxiety and 6 on depression. Score ranges from 0–3 for each item; total score ranges from 0 to 36, and a higher score indicates greater distress.	1
MFI, 1995 [46]	Multidimensional Fatigue Inventory	Measure fatigue in patients with cancer, categorized into five dimensions: general fatigue, physical fatigue, mental fatigue, reduced motivation, and reduced activity.	20 items, total score on each subscale ranges from 4 to 20, a higher score indicates a higher degree of fatigue.	1
CWS, 2010 [47]	Cancer Worry Scale	To assess concerns about developing cancer or developing cancer again and the impact of these concerns on daily functioning.	8 items, scores range from 1–4. Total scores range from 8 to 32 [13], with a score of 14 or higher being indicative of severe fear of cancer recurrence (FCR).	1
FCRI, 2009 [48]	Fear of Cancer Recurrence Inventory	To assess the multidimensional aspects of fear of cancer recurrence (FCR) on seven subscales: triggers, severity, psychological distress, coping strategies, functioning impairments, insight, reassurance.	42 items, scores range from 0 to 4. A total score can be obtained for each subscale and for the total scale, a higher score indicates higher levels of FCR.	1
IES, 1979 [49,50]	Impact of Event Scale	To assess the frequency of intrusive and avoidant phenomena during or after the traumatic experience of cancer.	15 items, divided into two dimensions: intrusion (seven items, scores range from 0–35) and avoidance (eight items, scores range from 0–40). A total score of 9—25 reflects moderate adaptation difficulties; a score higher than 26 indicates serious adaptation difficulties.	1
CIS-fatigue, 1994 [51]	Checklist Individual Strength-Fatigue Severity scale	To assess fatigue severity.	8 items for severity, scores range from 8 to 56. A score of 35 points or higher indicates severe fatigue.	1
SES, 1998 [52]	Self-Efficacy Scale	To measure the sense of control regarding fatigue.	7 items, four-point Likert scale, higher total scores are indicative for more self-efficacy.	1
FCS, 1998 [53]	Fatigue Catastrophizing Scale	To measure catastrophizing in response to fatigue.	10 items, five-point Likert scale, computing the mean of 10 items derives a total score. A higher total score indicates more catastrophizing.	1
MMAS, 2008 [54]	Morisky Medication Adherence Scale	To assess patient medication adherence.	8 items, summary score ranges from 0 to 8. Low adherence (score = 6), medium adherence (score > 6 and < 8), and high adherence (score 8). The study included in this review defined patients with score < 8 as nonadherent.	1
SSRS, 1994 [55]	Social Support Rating Scale	To measure support received in society, including 3 items for objective support, 4 items for subjective support and 3 items for support utilization.	10 items, total scores range from 12 to 66, with higher scores representing higher levels of social support.	1
CTCAE, 1999–2017 [56]	Common Terminology Criteria for Adverse Events	A standard classification and severity grading scale to report adverse events (AE).	Grades range from 1 to 5, with unique clinical descriptions of severity for each AE based on this general guideline: Grade 1—Mild Grade 2—Moderate Grade 3—Severe or medically significant but not immediately life-threatening Grade 4—Life-threatening consequences Grade 5—Death related to AE	75

#### 3.3.5. Specific HRQoL Issues

Two cross-sectional studies assessed specific HRQoL issues. Fear of recurrence or disease progression occurred in 52% of the GIST patients [22]. GIST patients with high levels of fear scored lower on global QoL and the subscales of role, emotional, cognitive, and social functioning. They also experienced higher levels of psychological distress and difficulty making plans for the future. Severe fatigue occurred in 30% of the GIST patients compared to 15% in the matched healthy controls [35]. Severely fatigued patients had a lower global QoL, increased impairment in all the functional domains, lower self-efficacy, and more distress. One study [34] assessed the effect of two online interventions on fatigue, distress, and HRQoL. Patients were randomized to an internet-delivered cognitive behavior therapy or online psychoeducation program. Both interventions led to a reduction in fatigue and distress and an increase in global QoL and all functioning domains of HRQoL.

#### 3.3.6. Qualitative Studies

A qualitative study [36] on 50 GIST patients concluded that patients shared common experiences during each stage of disease management. Patients felt a sense of crisis during diagnosis, followed by hope upon TKI therapy initiation. Over time, they came to adapt to their new lives with GIST while acknowledging that TKI therapy could have an impact on their daily lives. With each follow-up, patients confronted the uncertainty of becoming TKI resistant and the possible need to switch therapy. Disease progression and TKI switching caused patients to revert to crisis and restart their emotional journey. Another, more recent qualitative study [23] among 20 GIST patients with metastatic disease found that more than half of the patients experienced side effects that influenced their daily lives in negative and challenging ways, which urged them to adapt to ‘a new normal’. Apart from the well-known side effects of imatinib, patients also reported an increased need for sleep, cognitive challenges, reduced sexual desire, as well as poor stress tolerance around the intake of their GIST medication.

### 3.4. Adverse Events

Ninety-six studies with a prospective (*n* = 58) or retrospective (*n* = 38) design reported on adverse events. GIST patients were treated in various settings with different types of TKIs depending on their disease status. As adverse events differ between TKIs, study characteristics and main findings are summarised per type of TKI in Table 3. Studies used different versions of the Common Terminology Criteria for Adverse Events (CTCAE) to report and grade adverse events; in twenty studies, the used measure was not described. The characteristics of the CTCAE are described in Table 2.

#### 3.4.1. Imatinib

GIST patients received imatinib in different treatment settings; neo-adjuvant (*n* = 6), adjuvant (*n* = 13), or palliative setting (*n* = 37). In nine studies, patients in different treatment settings were combined, or the treatment setting was unknown. Patients were mainly treated with imatinib 400 mg once a day, but other doses such as 300 mg once a day, 600 mg once a day, or 400 mg twice a day were also prescribed. Median treatment durations varied from 3 days to 9 months in the neo-adjuvant setting, from 181 days to 5 years in the adjuvant setting, and from 2 months up to 8 years in a palliative setting. Hematological adverse events, including anemia, leukopenia, and neutropenia, were common. Furthermore, dermatitis, diarrhea, fatigue, muscle cramps or spasms, nausea, periorbital edema, peripheral edema, and rash were frequently reported non-hematological adverse events. One study in particular [78] compared adverse events before and after dose escalation to 800 mg/day in 133 patients with progression on imatinib 400 mg/day and concluded that anemia and fatigue were more likely to be worse after dose escalation. In six studies, GIST patients were treated with imatinib for a median duration of at least 3 years. In an international randomized controlled trial [18], 198 patients received 3 years of adjuvant imatinib. All patients experienced at least one adverse event, mostly anemia, periorbital edema, diarrhea, nausea, and muscle cramps. Another study [26] assessed if adjuvant imatinib for 5 years was tolerable and efficacious, in which 91 patients were treated for a median duration of 55.1 months, and 46 patients completed the intended 5 years of adjuvant treatment. The most common adverse events in this study were nausea, diarrhea, fatigue, periorbital edema, and muscle spasms. A study [29] in a real-world setting reported on 151 patients with unresectable or metastatic GIST treated for a median duration of 42.6 months. Out of the 151 patients, 148 patients experienced adverse events, mostly diarrhea, asthenia, and eyelid or periorbital edema. A retrospective multicenter study [92] aimed to gain insight into GIST patients with unresectable or metastatic disease responding long-term to imatinib. Of these 58 long-term responders treated with imatinib for 5.5 to 10.4 years, 15 patients experienced new emerging adverse events after ≥ 5 years of treatment, including anemia, fatigue, renal failure, diarrhea, edema, and muscle cramps.

#### 3.4.2. Sunitinib

Twenty-five studies reported on adverse events of sunitinib in patients with unresectable or metastatic GIST who progressed on or were intolerant to imatinib. In twenty-two studies, patients received sunitinib in a fractioned dose, mainly 50 mg a day in a 4-weeks-on-2-weeks-off schedule, and in eight studies, patients were treated with a continuous dose of 37.5 mg (or lower) once a day. The median treatment duration of sunitinib varied from 12.8 weeks to 60 weeks. Hematological adverse events were common, including anemia, thrombocytopenia, leukopenia, and neutropenia. Frequent non-hematological adverse events were nausea, vomiting, decreased appetite, abdominal pain, diarrhea, asthenia, fatigue, hypertension, liver dysfunction, hand–foot syndrome, rash, skin discoloration, and hair color change. In addition, mucositis and hypothyroidism were reported as grade 3 and 4 adverse events. Three studies [126,127,128] focused on the effect of sunitinib on the thyroid function. Respectively, 15 of the 42, 10 of the 24, and 2 of the 17 sunitinib treated GIST patients developed hypothyroidism. One study [136] in 75 metastatic GIST patients was conducted to determine the cardiovascular risk of sunitinib. In total, 35 patients developed hypertension, 8 patients suffered a cardiovascular event, and in 10 of the 36 patients, a left ventricular ejection fraction (LVEF) decline of ≥10% occurred. In a few studies, fractioned dosing was compared to continuous dosing of sunitinib. The overall profile of AEs was comparable, but the incidence of AEs in patients receiving continuous dosing was slightly higher [134,135]. A large treatment-use trial of 1124 sunitinib treaded GIST patients compared modified dosing schedules to the fractioned dosing schedule of 50 mg 4-weeks-on-2-weeks-off. Patients with modified dosing schedules experienced more AEs, both any grade and grade 3/4, but the number of patients that discontinued treatment was lower (26% vs 34%) [121].

#### 3.4.3. Regorafenib

Ten studies investigated patients with advanced or metastatic GIST treated with regorafenib, mostly after failure of imatinib and sunitinib. In eight studies, patients were treated with regorafenib for the first 3-weeks in a 4-week cycle, and in two studies, patients received a continuous dose once a day. The median treatment duration of regorafenib varied from 20 weeks to 60 weeks. The most common adverse event was hand–foot skin reaction (56–92%). Other frequently reported adverse events were diarrhea, fatigue, hoarseness, hypertension, and oral mucositis. One study [147] specifically investigated the incidence of regorafenib-associated hepatic toxicity, in which 5 of the 21 metastatic GIST patients developed (laboratory) hepatic toxicity. In a single-center retrospective study of 28 patients, toxicity and efficacy of regorafenib in fractioned dosing, 160 mg a day for the first 3 weeks of every 4-week cycle, was compared to continuous dosing of 120 mg a day. Despite the small numbers, the study concluded that continuous dosing was better tolerated with comparable efficacy [146].

#### 3.4.4. Ripretinib

One international randomized, placebo-controlled trial [33] reported adverse events in 85 GIST patients with advanced disease who received ripretinib as a fourth-line therapy. The most common grade 1 and 2 adverse events were alopecia, myalgia, nausea, fatigue, and hand–foot syndrome. Grade 3 or 4 adverse events were rare, but lipase increase, hypertension, and anemia of these grades were reported.

## 4. Discussion

Imatinib, sunitinib, and regorafenib were all approved for the treatment of GIST based on studies without any HRQoL data. In the two decades that GIST patients have been treated with TKIs, the number of studies addressing HRQoL is remarkably low. Available literature showed that HRQoL in patients responding to imatinib, regorafenib, and ripretinib remained stable, while most sunitinib-treated patients reported a decrease in HRQoL. Imatinib, sunitinib, and regorafenib are also registered for the treatment of other cancer types. Before its approval as a treatment for advanced and metastatic GIST, imatinib was approved for the treatment of chronic myeloid leukemia (CML). In imatinib-treated CML patients, physical functioning and well-being remained stable during 18 months of treatment [148]. Another study investigated whether patients with CML treated with long-term imatinib had a different HRQoL compared to their respective peers without cancer in the general population. The HRQoL of CML patients on imatinib age 60 years or older was comparable with that of their peers, while younger patients and women reported the largest HRQoL differences compared to their peers [149]. Most studies included in this review did not compare HRQoL of GIST patients to a normative population, but one study compared the prevalence and severity of fatigue with matched healthy controls. Severe levels of fatigue were found in 30% of the GIST patients compared with 15% in matched healthy controls and were associated with worse HRQoL. In CML patients treated with long-term imatinib, chronic fatigue was found to be the most important factor limiting HRQoL [150]. Sunitinib is also used in the treatment of metastatic renal cell carcinoma (RCC) and metastatic gastroenteropancreatic neuroendocrine tumors (GEP-NETs). Patients with metastatic RCC treated with sunitinib had a stable overall HRQoL during treatment, but their physical well-being worsened over time [151]. The health status of sunitinib-treated patients with GEP-NETs remained stable during the first six cycles (50 mg a day 4-weeks-on-2-weeks-off) of treatment [152]. These results differ from a study that assessed HRQoL in GIST patients, in which most patients reported a decrease in HRQoL; however, we need to take into account the small number of patients (*n* = 13), and the timing of HRQoL assessment being unclear. An explanation might be that RCC patients received sunitinib as first-line treatment and therefore have no comparison with other TKI therapies, while GIST patients had prior treatment with imatinib, which is well tolerated in general. A frequently reported side effect in sunitinib-treated patients is hand–foot skin reaction, which negatively impacts HRQoL [153]. Regorafenib is a third-line treatment for GIST patients but is also registered as a treatment for hepatocellular carcinoma (HCC) and metastatic colorectal cancer (mCRC). Hofheinz et al. pooled HRQoL data of patients treated with regorafenib in four trials, including GIST, HCC, and mCRC patients [154]. Across all tumor types, regorafenib significantly delayed the patient’s first clinical important deterioration in HRQoL score, with the median time to HRQoL deterioration ranging from 16–24 weeks for regorafenib compared to 8–12 weeks for placebo.

Studies included in this review showed that nearly all patients treated with a TKI experienced at least one adverse event, mostly mild to moderate. Severe adverse events were uncommon but did occur. TKI therapies had different side effects related to agent, dose, treatment duration, age, and ethnicity. A review of the safety data of imatinib in CML and GIST patients showed that imatinib treatment led to similar side effects in both diseases [155]. However, severe nausea and diarrhea were more frequent in GIST than in CML patients when treated with the same dose, and this may be due to the origin of GIST and the fact that GIST patients often had previous gastrointestinal surgery. In line with the studies on GIST patients receiving long-term imatinib, imatinib in CML patients was not associated with unacceptable cumulative or late toxic effects [156]. Although imatinib is generally well tolerated, nausea, edema, and fatigue are the main reasons for dose reductions in imatinib-treated GIST patients [72,76]. Furthermore, in daily clinical practice, dose reductions are applied in case of intolerable side effects to continue treatment and maintain a patient’s HRQoL. Sunitinib for GIST can be prescribed in two different schedules, a fractioned dosing schedule of 50 mg a day for 4 weeks followed by 2 weeks off or a continuous dosing schedule of 37.5 mg a day. As previously discussed in the results, patients receiving modified dosing schedules experienced more AEs, but fewer patients discontinued treatment, resulting in a longer median overall survival of 23.5 months compared to 11.1 months in patients receiving a fractioned dosing schedule of 50 mg 4-weeks-on-2-weeks-off [121]. This study underlines the importance of appropriate dose adjustments, resulting in a tolerable prolonged treatment with beneficial clinical outcomes. Alternative dosing schedules of sunitinib were also assessed in patients with RCC. Results suggest that a 2-weeks-on-1-week-off schedule is less toxic with similar efficacy, while there was no benefit in safety or efficacy for continuous dosing compared to a 4-weeks-on-2-weeks-off schedule [157]. The toxicity profiles of regorafenib in the treatment of GIST, HCC, and mCRC patients were comparable [158]. Among GIST patients, hand–foot skin reaction (HFSR) was the most common adverse event. The incidence of HFSR varied significantly per tumor type and was 60.2% for GIST, 50.0% for HCC, and 46.6% for mCRC [159]. There is no evidence for a relationship between the incidence of HFSR and previous TKI use or duration of TKI use, the exact molecular mechanisms behind the increased incidence are poorly understood, and the occurrence of HFSR seems rather dose-related. Clinicians mostly prescribe regorafenib in a fractioned dosing schedule of 160 mg a day in a 3-weeks-on-1-week-off schedule, which often leads to unacceptable toxicity resulting in lowering the dose, intermittent drug withdrawal, or complete drug withdrawal. A meta-analysis of studies focusing on regorafenib-associated AEs reported a significant correlation between the occurrence of adverse events and the recommended dose of 160 mg (3 weeks on-1 week off), while no significant correlation was found at a dose of 120 mg with a similar schedule [160]. Data on the optimal dosing of regorafenib are limited, but these results suggest that a dose of 120 mg might be a better fit. Only one small single-center study in GIST patients compared different dosing schedules of regorafenib and concluded that continuous dosing of 120 mg daily was better tolerated with comparable efficacy [146]. Another study pointed out that dosing of regorafenib and toxicity management is critical, as the median duration of treatment was longer [145], which may lead to a durable clinical benefit. Furthermore, less toxicity during treatment will probably result in a better HRQoL, as patients experience fewer physical complaints and uncertainty of needing to interrupt or stop treatment due to intolerable side effects.

The majority of studies included in this review used a physician-reported measure, mostly the CTCAE, to rate and grade side effects. The CTCAE might not be the appropriate measure to report TKI-related side effects due to the subacute and persisting nature of these side effects. The needed treatment adjustments during TKI therapy in clinical practice underscore the limitations of the CTCAE as a measure for AE reporting. The short and severe toxicities due to conventional chemotherapy differ from the daily and long-lasting lower grade toxicities of TKI therapy [161]. In addition, previous research has shown a gap between physician-reported and patient-reported outcomes. Physicians tend to underreport symptoms, as patients report symptoms earlier and more frequently with worse symptom severity than physicians [162,163]. This phenomenon is also observed in the treatment with TKIs; physicians underestimated the severity of long-term side effects of imatinib in CML patients, in particular for muscle cramps and musculoskeletal pain [164]. In order to create a more complete overview of side effects and symptoms, the patient’s perspective is needed. With the continuing improvements in cancer treatment, the use of PROMs in cancer research is considered more important, for example, the use of the PRO-CTCAE to capture patient-reported side effects. It becomes relevant to not only assess treatment effectiveness in terms of objective clinical outcomes (e.g., response, recurrence, and survival) but also in terms of patient-reported outcomes, to determine the net clinical benefit of treatments. This suggests that time to HRQoL improvement and time to sustained HRQoL improvement are potentially important outcomes [165].

In this review, only 13 of the 104 included studies used PROMs. The EORTC QLQ-C30 was the most frequently used PROM, while FACT-G, EQ-5D, and SF-36 were sporadically used. These PROMS are population-generic (e.g., SF-36, EQ-5D) or cancer-generic (e.g., EORTC QLQ-C30, FACT-G) and have the disadvantage that they do not cover all relevant issues for GIST patients on TKI therapy. For instance, (periorbital) edema, muscle pain and cramps, and hand–foot skin reaction are not included in these PROMs, indicating that they may lack content validity. On the other hand, the MDASI-GIST, a PROM that was developed particularly for GIST, was never used. An explanation could be that this questionnaire focuses on nine imatinib-induced side effects and therefore does not cover side effects for other lines of TKI therapy. Recently, the EORTC Symptom Based Questionnaire (EORTC-SBQ) was developed for patients receiving targeted therapy [166]. In the EORTC-SBQ, many TKI-related side effects are included, i.e., periorbital edema, peripheral edema, muscle cramps or pains, pain or soreness in the mouth, and skin problems. The EORTC is also developing a survivorship core questionnaire (EORTC-SURV100) to assess the late effects of cancer diagnosis, treatment, and HRQoL in cancer survivors [167]. Both questionnaires can be used in future HRQoL research in combination with a still to be developed questionnaire addressing GIST-specific issues, such as problems because of changed appearance or the feeling that the impact of having a GIST and the side effects of treatment are not understood by friends or family. By combining these different PROMs, a new measurement strategy is applied to cover all relevant issues of GIST patients on (long-term) TKI therapy.

The strength of this review is that it is the first to provide an overview of the available literature on HRQoL and the side effects of different TKIs used in the treatment of GIST patients. With this review, awareness of potential side effects and their impact on HRQoL is raised. Both health care professionals and GIST patients are provided with information that can have important implications for patients’ HRQoL, (shared) decision-making, treatment strategies, and clinical outcomes. Because avapritinib and ripretinib were approved recently, this review does not include studies on avapritinib and only one study on ripretinib. In 2021, the results of the VOYAGER, a phase III study comparing avapritinib with regorafenib as third-line or later treatment in patients with unresectable or metastatic GIST, were published. There was no significant difference in median progression-free survival or treatment-related adverse events between both treatments, but the type of adverse events differ with avapritinib inducing more cognitive effects (25.9% vs 3.8%). Due to the low number of studies investigating HRQoL and incorporating PROMs, a limited overview of HRQoL issues for GIST patients on TKI therapy is presented. Furthermore, presented HRQoL data need to be interpreted with caution, as assessment of HRQoL was often conducted in small samples, after a relatively short duration of treatment, or stopped after disease progression or cross-over. Therefore, important aspects of long-term TKI treatment or treatment discontinuation could be missing.

## 5. Conclusions

In conclusion, this review showed that most TKI treated GIST patients experience side effects, mostly mild to moderate, which did not seem to affect overall HRQoL. However, in daily clinical practice, side effects and their impact on the daily lives of patients are the main reason for dose reductions, dose interruptions, and schedule modifications. Treatment adjustments are needed in order to maintain a patient’s HRQoL, risking worse clinical outcomes, but pre-emptive toxicity management can result in a longer duration of therapy, hence the importance of HRQoL. On the one hand, this suggests that the reported side effects were underestimated, as most studies used the CTCAE, a physician-reported measure, to rate and grade side effects. Apart from the fact that the CTCAE might not be the appropriate measure to report TKI-related side effects, previous research has shown a gap between physician-reported and patient-reported outcomes. On the other hand, using cancer-generic PROMs might not capture all relevant issues that determine a GIST patient’s HRQoL. Therefore, a new measurement strategy should be applied to detect, with more sensitivity, patient-reported side effects, symptoms, and HRQoL issues relevant to GIST patients on TKI therapy.

## Figures and Tables

**Figure 1 cancers-14-01832-f001:**
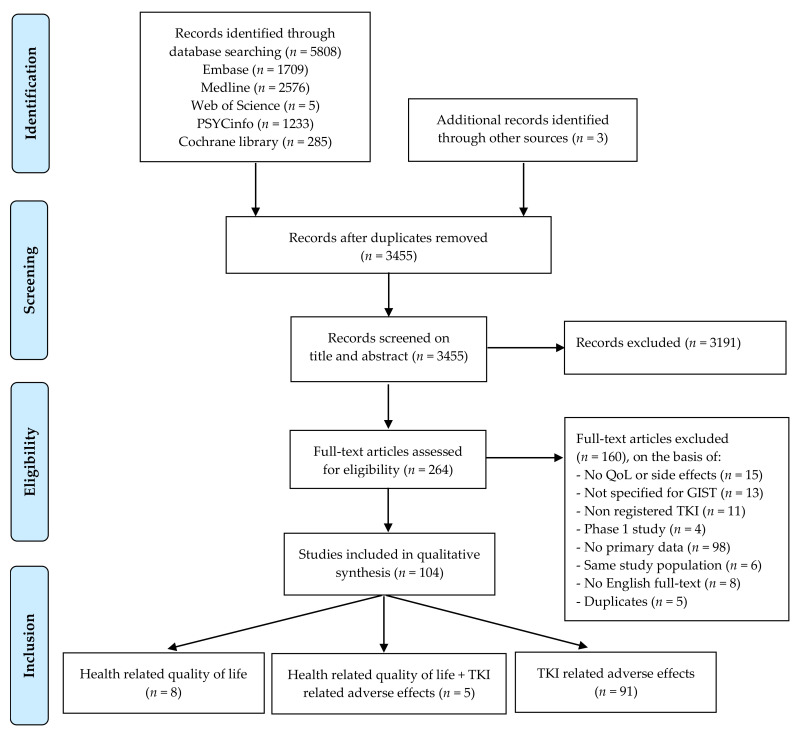
Flow chart of the selection procedure.

**Table 3 cancers-14-01832-t003:** Studies reporting on adverse events.

Imatinib in Neo-Adjuvant Setting							
Author, Year, Country (Ref)	Design	Aim	Patients Characteristics (Number of Patients, Gender Male %, Age Range)	Treatment/Intervention	Outcome Measure	Results	Quality Score
Kurokawa, 2017, Japan [57]	Prospective, international, multicenter, phase II	To investigate the efficacy and safety of neoadjuvant imatinib for 6–9 months	53 patients with large (≥10 cm) gastric GISTs, 45% male, median age of 69 years (range 43–79)	Imatinib 400mg/day for a median duration of 26 weeks (range 1.7–39.6)	CTCAE v3.0 Grade 1, 2, 3, 4 and all grade AEs	The most common reported AEs of all grade were anemia (94%), neutropenia (72%), periorbital edema (72%), leukopenia (51%) and rash (45%). Most frequently reported grade 3 AEs were rash (9%), neutropenia (4%) and leukopenia (4%). Grade 4 AEs occurred in 3 patients, including neutropenia (4%) and CNS ischemia (2%).	4 *
Doyon, 2012, Canada [58]	Prospective, multicenter	To evaluate the optimal neoadjuvant imatinib duration to reduce the morbidity of surgery and increase the possibility of resection completeness	14 patient with locally advanced GIST, 86% male, mean age 64 years (range 39–84)	Imatinib 400 mg/day (*n* = 7) or imatinib 600 mg/day (*n* = 7) for a median duration of 9 months (range 2–12)	CTCAE v2.0 Grade 3 AEs	Grade 3 nausea was reported in 1 patient.	4 *
Ashraf, 2011, India [59]	Prospective, single center	To study the usefulness of imatinib in downstaging or downsizing locally advanced GISTs prior to surgery	19 patients with locally advanced GIST without metastasis, 16% male, mean age 38 years (range 26–74)	Imatinib 400 mg/day for a median duration of 140 days (range 84 to 168)	CTCAE v2.0 No grades	Reported AEs were edema (26%), anorexia (16%), constitutional symptoms (11%) and severe neutropenia (5%).	4 *
Tielen, 2013, The Netherlands [60]	Retrospective, multicenter	To analyze the outcome of multimodality treatment for rectal GIST	32 patients with rectal GIST, 69% male, median age of 60 years (range 45–80)	22 patients received neo-adjuvant imatinib 400 mg/day for a median duration of 9 months (range 2–53)	Unknown No grades	AEs were experienced in 12 patients including periorbital edema (23%), fatigue (14%) and skin toxicity (14%). One patient developed severe skin toxicity and discontinued imatinib.	4 *
**Imatinib in neo-adjuvant + adjuvant setting**							
**Author, Year, Country (Ref)**	**Design**	**Aim**	**Patients Characteristics (Number of Patients, Gender Male %, Age Range)**	**Treatment/Intervention**	**Outcome Measure**	**Results**	**Quality Score**
Eisenberg, 2009, USA [61]	Prospective, multicenter, phase II	To determine the outcome and toxicity of imatinib given as a neoadjuvant agent prior to resection of intermediate/high risk primary GIST or metastatic/recurrent GIST	52 patients with primary ≥ 5 cm or recurrent/metastatic ≥ 2 cm operable GIST, 54% male, median age 58 years (range 24–84)	Imatinib 600 mg/day prior to surgery for a median duration of 65 days and continued for 2 years as a postoperative adjuvant	CTCAE v2.0 Grade 4 and 5 AEs	The most frequent reported grade 4 AE during neo-adjuvant imatinib treatment was neutropenia (8%), the only grade 5 AE that occurred was pneumonitis (2%). The most frequent grade 4 AE during adjuvant imatinib treatment was anorexia (4%), reported grade 5 AEs were constitutional symptoms (2%) and hemorrhagic stroke (2%).	4 *
McAuliffe, 2009, USA [62]	Prospective, single center, randomized, phase II	To assess the safety and efficacy of preoperative and postoperative imatinib for the treatment of GIST	19 patients undergoing surgical resection of kit-expressing GIST (≥1 cm), 58% male, mean age of 59 years	Patients received 3 (*n* = 7), 5 (*n* = 6), or 7 (*n* = 6) days of preoperative imatinib 300 mg twice a day and postoperative imatinib 600 mg/day for 2 years, 8 patients completed the intended 2 year of treatment	CTCAE v3.0 Grade 3 and 4 AEs	Preoperative 1 case of grade 4 nausea/vomiting (5%) and 1 case of grade 4 dehydration (5%) were reported. During the postoperative imatinib treatment grade 3 AEs were edema (5%), nausea (5%), arrhythmia (5%), hypocalcemia (5%), dizziness (5%), memory impairment (5%), fatigue (5%), anorexia (5%) and vaginal bleeding (5%). One patient experienced grade 4 anemia (5%).	2 *
**Imatinib in adjuvant setting**							
**Author, Year, Country (Ref)**	**Design**	**Aim**	**Patients Characteristics (Number of Patients, Gender Male %, Age Range)**	**Treatment/Intervention**	**Outcome Measure**	**Results**	**Quality Score**
Casali, 2015, Italy [63]	Prospective, international, multicenter, randomized, phase III	To assess the efficacy of adjuvant imatinib	908 patients with localized GIST with an intermediate or high risk of relapse after R0 or R1 surgery, 51% male, median age 59 years (range 18–89)	449 patients received imatinib 400 mg/day for an unknown duration, 339 patients completed the intended 2 years of treatment	CTCAE v3.0 Grade 3/4 AEs occurring in ≥2.5%	The main grade 3/4 AEs were neutropenia (6.2%), weight loss or gain (3.3%), infections (3.1%) and ALT increase 2.8%.	3 *
DeMatteo, 2009, USA [64]	Prospective, international, multicenter, randomized, phase III	To assess if adjuvant treatment with imatinib would improve RFS compared to placebo treatment	713 patients with complete gross resection of a primary GIST of ≥3 cm, 51% male, median age 58 years (range 18–91)	337 patients received imatinib 400 mg/day for an unknown duration, 240 patients completed the intended 1 year of treatment	CTCAE v3.0 Grade 1, 2, 3, and 4 AEs	The most common grade 1/2 AEs were periorbital or peripheral edema (75%), fatigue (41%), diarrhea (28%), nausea (27%) and dermatitis (20%). The most frequent grade 3 AEs were abdominal pain (3%), dermatitis (3%), diarrhea (2%), nausea (2%) and vomiting (2%). Grade 4 AEs were reported in 15 patients, most commonly neutropenia (1%), fatigue (<1%), increased ALT (<1%) or AST (<1%). Grade 5 AEs occurred in 3 patients.	4 *
Joensuu, 2012, Finland [18]	Prospective, international, multicenter, randomized, phase III	To investigate the role of imatinib administration duration as adjuvant treatment	199 patients with high estimated risk for GIST recurrence after surgery, 52% male, median age of 62 years (range 23–84)	Imatinib 400 mg/day for 12 months	CTCAE v2.0 The 10 most frequent AEs of any grade and grade 3/4	All 194 patients experienced at least one AE of any grade, most commonly anemia (72%), periorbital edema (59%), fatigue (48%), nausea (45%) and diarrhea (44%). Grade 3/4 AEs occurred in 39 patients, mostly leukopenia (2%), nausea (2%) and fatigue (1%).	4 *
			198 patients with high estimated risk for GIST recurrence after surgery, 49% male, median age of 60 years (range 22–81)	Imatinib 400 mg/day for 36 months	CTCAE v2.0 The 10 most frequent AEs of any grade and grade 3/4	All 198 patients experienced at least one AE of any grade, most commonly anemia (80%), periorbital edema (74%), diarrhea (54%), nausea (51%) and muscle cramps (49%). Grade 3/4 AEs occurred in 65 patients, mostly leukopenia (3%), diarrhea (2%), periorbital edema (1%), muscle cramps (1%) and leg edema (1%).	
DeMatteo, 2013, USA [65]	Prospective, multicenter, phase II	To conduct the first adjuvant trial of imatinib for treatment of GIST	106 patients after complete gross resection of a primary GIST with a high risk of recurrence, 57% male, median age 58 years (range 19–79)	Imatinib 400 mg/day for an unknown duration, 88 patients completed the intended 1 year of treatment	CTCAE v3.0 Any grade, grade 1, 2 and 3 AEs	The most frequent reported AEs of any grade were edema (57%), fatigue (52%), diarrhea (51%), nausea (49%), and dermatitis (36%). The most common grade 3 AEs were nausea (3%), ALT or AST elevation (3%), dermatitis (3%), neutropenia (2%) and abdominal pain (2%). No grade 4 or 5 AEs were reported.	5 *
Raut, 2018, USA [26]	Prospective, multicenter, phase II	To determine whether adjuvant treatment with imatinib for 5 years is tolerable and efficacious	91 patients with intermediate or high risk of recurrence after resection of primary GIST, 53% male, median age of 60 years (range 30–90)	Imatinib 400 mg/day for a median duration of 55.1 months (range 0.5–60.6), 46 patients completed the intended 5 years of treatment	CTCAE V4.03 Any grade and grade 3/4 AEs occurring in ≥10%	All 91 patients experienced at least 1 AE. The most common AEs of any grade were nausea (62%), diarrhea (50%), fatigue (37%), periorbital edema (33%) and muscle spasms (32%). Grade 3/4 AEs were identified in 17 patients, mostly diarrhea (2%) and abdominal pain (2%).	3 *
Kanda, 2013, Japan [66]	Prospective, multicenter, phase II	To determine the efficacy and safety of imatinib adjuvant therapy for Japanese GIST patients	64 patients with primary high-risk GIST after complete resection, 64% male, median age of 59 years (range 27–74)	Imatinib 400 mg/day for an unknown duration, 49 patients completed the intended 1 year of treatment	CTCAE v2.0 Any grade, grade 1, 2, 3 and 4 AEs occurring in ≥ 10%	All patients reported at least one AE of any grade, most frequently eyelid edema (48%), neutropenia (41%), leukopenia (39%), nausea (39%), rash (38%) and peripheral edema (38%). Grade 3 AEs occurred in 17 patients, most commonly neutropenia (13%), leukopenia (5%), rash (3%) and lymphopenia (3%). A total of 5 patients experienced grade 4 AEs, including 1 case of neutropenia.	4 *
Kang, 2013, Korea [67]	Prospective, multicenter, phase II	To evaluate the efficacy and safety of 2-year adjuvant imatinib	47 patients at high risk of recurrence after complete resection of localized GIST with KIT exon 11 mutation, 51% male, median age of 57 years (range 36–74)	Imatinib 400 mg/day in 1 month cycles for a median duration of 24 cycles (range 1–24)	CTCAE v3.0 Grade 1, 2, 3 and 4 AEs	The most commonly reported grade 1/2 AEs were edema (89%), anemia (81%), anorexia (57%), diarrhea (53%) and asthenia (53%). The most frequently reported grade 3 AEs were neutropenia (23%), dermatitis (9%), leukopenia (6%), anemia (4%), increased ALT (4%) and anorexia (4%). There were 2 cases of grade 4 neutropenia (4%) and 1 case of grade 4 leukopenia (2%). There were no treatment-related deaths.	4 *
Reichardt, 2019, Germany [68]	Prospective, international, multicenter	To evaluate the safety and tolerability of imatinib in an adjuvant setting	300 patients after complete resection of the primary GIST with a high or intermediate risk of relapse, 54% male, median age of 60 years (range 19–89)	Imatinib 400 mg/day for a median duration of 181 days (range 9–420)	CTCAE v3.0 Any grade, grade 1/2 and 3/4 occurring in ≥10%	Data of 28 patients were missing. A total of 272 patients experienced at least one AE of any grade, most frequently nausea (34%), diarrhea (33%), periorbital edema (21%), muscle spasms (16%) and peripheral edema (15%). Grade 3/4 AEs occurred in 45 patients including rash (1%), abdominal pain (1%) and diarrhea (1%).	4 *
Jiang, 2011, China [69]	Prospective, single center	To evaluate the efficacy and safety of extending imatinib adjuvant therapy for 5 years	90 patients with high-risk GISTs after R0 resection, 60% male, mean age of 54 years (range 17–80)	35 patients received imatinib 400 mg/day for a median duration of 33.8 months (range 3–60) and	CTCAE v3.0 Grade 1/2 and 3/4 AEs occurring in ≥10%	The most common grade 1/2 AEs were edema (71%), nausea (37%), fatigue (34%), leucopenia (31%) and anemia (29%). Most frequently reported grade 3/4 AEs included leucopenia (9%), edema (9%), fatigue (9%) and anemia (6%). No treatment-related deaths occurred.	4 *
Rutkowski, 2020, Poland [70]	Retrospective, multicenter, observational	To analyze the real world results of adjuvant imatinib treatment	107 patients with high risk GIST, 49% male, median age of 59 years (range 33–88)	Imatinib 400 mg/day for a median duration of 901 days	CTCAE v4.0 The 10 most common AEs of any grade and grade 3/4	The most common AEs of any grade were fluid retention (22%), skin toxicity (11%), nausea (7%), abdominal pain (6%) and fatigue (5%). The most frequently reported grade 3/4 AEs were fluid retention (2%), anemia (2%) and neutropenia (2%).	4 *
Wu, 2018, China [71]	Retrospective, multicenter	To determine whether imatinib adjuvant treatment improved recurrence-free survival	192 patients who underwent complete resection (R0) of localized primary GIST with intermediate recurrence risk, 50% male, median age of 55 years (range 52–58)	Data reported of 59 patients receiving adjuvant imatinib 400 mg/day for 1 (*n* = 1), 2 (*n* = 25) or 3 (*n* = 26) years	CTCAE v3.0 Grade 1/2 and grade 3/4 AEs	The most common grade 1/2 AEs were edema (73%), neutropenia (21%), fatigue (15%), nausea (10%) and skin rash (9%). The only grade 3/4 AEs reported were neutropenia (3%) and skin rash (1%).	4 *
**Imatinib in palliative setting**							
**Author, Year, Country (Ref)**	**Design**	**Aim**	**Patients Characteristics (Number of Patients, Gender Male %, Age Range)**	**Treatment/Intervention**	**Outcome Measure**	**Results**	**Quality Score**
Blanke, 2008, USA [72]	Prospective, international, multicenter, randomized, phase III	To compare the progression-free and overall survival rates for conventional dose imatinib versus higher doses	345 patients with advanced GISTs, 54% male, median age 61 years (range 18–87)	Imatinib 400 mg once a day for an unknown duration	CTCAE v2.0 Grade 3, 4 and 5 AE categories	147 grade 3/4 AEs were reported, most common categories were blood/bone marrow (20%), pain (11%) and gastrointestinal (9%). A total of 2 patients had grade 5 AEs, including 1 case of blood/bone marrow.	4 *
			349 patients with advanced GISTs, 54% male, median age 61 years, (range 18–94)	Imatinib 400 mg twice daily for an unknown duration	CTCAE v2.0 Grade 3, 4 and 5 AE categories	210 grade 3/4 AEs were reported, most common categories were blood/bone marrow (27%), gastrointestinal (16%), cardiac toxicity (14%) and pain (12%). A total of 9 patients experienced grade 5 AEs, most frequent hemorrhage (*n* = 4).	
Blay, 2015, France [73]	Prospective, international, multicenter, randomized, phase III	To test the efficacy and safety of nilotinib versus imatinib as first-line therapy	644 patients with unresectable or metastatic GIST, who had received no prior systemic therapy for GIST or had experienced a recurrence ≥6 months after stopping adjuvant imatinib, 57% male, median age 59 years (range 18–88)	316 patients received imatinib 400 mg/day for a median duration of 14.9 months (range 0.4–37.0)	CTCAE v3.0 Grade 1/2 AEs occurring ≥10%, grade 3 and 4 AEs	293 patients reported AEs. The most frequently reported grade 1/2 AEs were nausea (32%), diarrhea (30%), peripheral edema (21%), fatigue (20%), vomiting (19%) and periorbital edema (18%). The most common grade 3 AEs were hypophosphatemia (6%), abdominal pain (4%), lipase increase (3%), anemia (3%) and neutropenia (3%). A total of 45 grade 4 AEs were reported, of which anemia (*n* = 7) was the most frequent.	4 *
Blay, 2007, France [28]	Prospective, multicenter, randomized, phase III	To determine whether interruption of imatinib is feasible in advanced GIST patients with controlled disease after 1 of imatinib	182 advanced GIST patients of which 98 patients in response or with stable disease under imatinib at 1 year of follow-up, 59% male, median age of 62 years (range 27–87)	Imatinib 400 mg/day for 12 months	Unknown The most common grade 3/4 AEs	47 of the 182 patients had at least one grade 3/4 AE, most frequently neutropenia (6%), asthenia (3%) and rash (3%).	2 *
Kang, 2013, Korea [74]	Prospective, single center, randomized, phase III	To determine the efficacy and safety of imatinib	81 patients with metastatic and/or unresectable GIST with prior benefit from imatinib and subsequent progression on at least imatinib and sunitinib, 68% male, median age of 59 years (IQR 52–67)	41 patients received imatinib 400 mg/day for an unknown duration	CTCAE v3.0 All grades AEs occurring in ≥ 10% and grade 3/4 AEs	All imatinib-treated patients experienced at least one AE of all grades, most frequently anemia (66%), edema (44%), fatigue (37%), anorexia (34%) and nausea (32%). A total of 39 patients experienced grade 3/4 AEs, most commonly anemia (29%), fatigue (10%), and hyperbilirubinemia (7%).	4 *
Reichardt, 2012, Germany [75]	Prospective, international, multicenter, randomized, phase III	To investigate the efficacy of nilotinib versus best supportive care with or without a TKI	248 patients with advanced GIST following failure of prior imatinib and sunitinib, 60% male, mean age of 58 years (range 18–83)	54 patients received imatinib 669.5 mg/day for a duration of 57.5 days	CTCAE v3.0 Any grade AEs occurring in ≥10% and grade 3/4 AEs occurring in >1%	52 patients experienced at least one AE of any grade, most frequently nausea (54%), peripheral edema (43%), vomiting (41%), anemia (37%) and anorexia (28%). Grade 3/4 AEs occurred in 5 patients, including anemia (7%), septic shock (2%) and gastrointestinal hemorrhage (2%).	4 *
Verweij, 2004, The Netherlands [76]	Prospective, international, multicenter, randomized	To assess dose dependency of response and progression-free survival with imatinib	473 patients with advanced or metastatic GIST, 60% male, median age 59 years (range 49–67)	470 patients received imatinib 400 mg/day for an unknown duration	CTCAE v2.0 Grade 1, 2, 3 and 4 AES	465 patients had at least one AE of any grade. The most common grade 1/2 AEs were anemia (82%), edema (69%), fatigue (62%), nausea (46%) and diarrhea (46%). The most frequent grade 3 AEs were fatigue (6%), anemia (6%), granulocytopenia (4%) and pleuritic pain (4%). A total of 29 patients had grade 4 AEs, most frequently granulocytopenia (3%) and anemia (1%).	4 *
			473 patients with advanced or metastatic GIST, 61% male, median age 60 years (range 49–68)	472 patients received imatinib 400 mg twice a day for an unknown duration	CTCAE v2.0 Grade 1, 2, 3 and 4 AES	468 patients had at least one AE of any grade. The most common grade 1/2 AEs were anemia (81%), edema (78%), fatigue (68%), nausea (57%) and diarrhea (51%). The most common grade 3 AEs were anemia (12%), fatigue (11%), edema (9%), bleeding (8%). A total of 36 patients had grade 4 AEs, most frequently anemia (5%), granulocytopenia (2%) and bleeding (2%).	
Verweij, 2007, The Netherlands [77]	Prospective, international, multicenter, randomized	To explore the database of the large EORTC-ISG-AGITGl study for imatinib induced cardiac toxicity	946 patients with advanced or metastatic GIST, 61% male, median age 59 years (range 49–68)	470 patients received imatinib 400 mg/day and 472 patients received imatinib 400 mg twice a day for a median duration of 24 months	CTCAE v2.0 Only cardiovascular system associated AEs	In 2 patients (0.2%) a possible cardiotoxic effect of imatinib could not fully be excluded.	4 *
Zalcberg, 2005, Australia [78]	Prospective, international, multicenter	To evaluate the feasibility, safety and efficacy of crossing over to the higher dose of imatinib at the time of progression	133 patients with progression on imatinib 400 mg/day in the EORTCISG-AGITG study, 65% male, median age of 59 years (range 20–85)	Imatinib 800 mg/day for a median duration of 112 days (range 83–154)	CTCAE v2.0 AEs after versus before cross-over at 60 days of follow-up	AEs after cross-over were compared to the same AEs observed in the same patient before cross-over. Anemia and fatigue were significantly more likely to be worse after cross-over. The most frequently reported new grade 3/4 AEs were anemia (13%), fatigue (8%) and edema (5%).	4 *
Demetri, 2002, USA [79]	Prospective, international, multicenter, randomized, phase II	To test the efficacy and safety of imatinib	147 patients with an unresectable or metastatic GIST, 56% male, median age of 54 years (range 18–83)	73 patients received imatinib 400 mg/day and	CTCAE v2.0 Any grade and grade 3/4 AEs occurring in ≥ 5%	71 patients experienced at least one AE of any grade, most commonly nausea (51%), periorbital edema (45%), diarrhea (40%), myalgia (37%) and fatigue (30%). A total of 15 patients reported grade 3/4 AEs, including neutropenia (7%), hemorrhage (4%), rash (3%), abnormal liver function results (3%) and leukopenia (3%).	3 *
				74 patients received imatinib 600 mg/day for an unknown duration	CTCAE v2.0 Any grade and grade 3/4 AEs occurring in ≥ 5%	73 patients experienced at least one AE of any grade, most frequently nausea (54%), diarrhea (50%), periorbital edema (50%), myalgia (42%) and fatigue (39%). Grade 3/4 AEs were reported in 16 patients, including hemorrhage (5%), diarrhea (3%), rash (3%), anemia (3%), neutropenia (3%) and abnormal liver function results (3%).	
Ryu, 2009, Korea [80]	Prospective, multicenter, phase II	To evaluate the efficacy and safety of imatinib and assess KIT and PDGFRA gene mutation status in Korean patients	47 patients with metastatic or unresectable KIT positive GIST, 57% male, median age of 57 years (range 31–81)	Imatinib 400mg/day for un unknown duration	CTCAE v2.0 Grade 1, 2, 3, 4 AEs	The most common grade 1/2 AEs were leukopenia (87%), anemia (81%), facial edema (53%), diarrhea (53%) and peripheral edema (53%). The most frequently reported grade 3 AEs were neutropenia (21%), anemia (17%), leukopenia (4%) and abdominal pain (4%). There were 2 grade 4 AEs, including anemia (2%) and neutropenia (2%).	4 *
Nishida, 2008, Japan [81]	Prospective, multicenter, phase II	To assess the efficacy and safety of imatinib in Japanese patients	74 patients with advanced GIST, 65% male, median age of 56 years (range 24–74)	28 patients received imatinib 400 mg/day and 46 patients received 600 mg/day for an unknown duration	CTCAE v2.0 Any grade and grade 3/4 AEs	400 mg: All patients experienced at least one AE of any grade, most frequently nausea (75%), diarrhea (71%), limb edema (68%), facial edema (57%), vomiting (57%) and dermatitis (54%). A total of 13 patients had grade 3/4 AEs including anemia (18%), and neutropenia (11%). 600 mg: All patients experienced at least one AE of any grade, most frequently nausea (80%), diarrhea (70%), dermatitis (67%), facial edema (63%), eyelid edema (52%), vomiting (52%), lower limb edema (52%) and muscle cramps (52%). A total of 27 patient had grade 3/4 AEs, most commonly neutropenia (28%), anemia (17%) and dermatitis (11%).	4 *
Schlemmer, 2011, Germany [82]	Prospective, multicenter, phase II	To open access to imatinib and assess the efficacy, safety and tolerability of imatinib	95 patients with unresectable or metastatic GIST, 56% male, median age of 59 years (range 18–80)	Imatinib 400 mg/day for an unknown duration	Unknown Any grade and for non-hematological AEs grade 1/2 and 3/4	70 patients experienced at least 1 AE, most frequently nausea (28%), peripheral edema (24%), eyelid edema (24%), diarrhea (21%) and muscle cramps (16%). Grade 3/4 AEs were uncommon and included nausea (1%), edema (1%), vomiting (1%), diarrhea (1%) and headache (1%).	4 *
Bouche, 2018, France [29]	Prospective, multicenter, observational	To describe the profile of treated patients, the prescription patterns and the impact of treatment on population health in a real-world setting	151 patients with unresectable or metastatic KIT-positive GIST, 58% male, median age of 60 years (range 21–86)	Imatinib 200–800 mg/day for a median duration of 42.6 months (range 4.9–86.7)	AEs occurring in ≥ 10 patients and serious AEs	148 patients reported AEs, most frequently diarrhea (39%), asthenia (39%), eyelid or periorbital edema (32%), abdominal pain (23%) and anemia (21%). A total of 8 of the 126 reported serious AEs were possibly related to imatinib, most frequently gastrointestinal disorders (*n* = 3).	3 *
Prenen, 2006, Belgium [83]	Prospective, multicenter	To evaluate the tumor response in GIST patients treated with imatinib and to assess its safety	57 patients with unresectable or metastatic GIST, 60% male, median age of 65 years (range 29–91)	Imatinib 400 mg/day for a median duration of 208 days (range 12–391)	Unknown	The main AEs were skin rash (58%), diarrhea (54%), asthenia (49%), nausea (40%) and periorbital edema (39%).	4 *
Rutkowski, 2018, Poland [84]	Prospective, mulicenter, observational	To analyze the treatment results of advanced GIST in the largest, homogenous series of older patients	656 patients (<70 years n= 517, ≥70 years *n* = 139) with metastatic/unresectable GIST, 55% male, median age of 59 years (range 15–89)	Imatinib 400 mg/day for an unknown duration	CTCAE v4.0 The 8 most common imatinib-related AEs of any grade and grade 3/4	Patients < 70 years old: 318 patients experienced at least one AE of any grade, most frequently fluid retention (52%), anemia (44%), neutropenia (43%), nausea (32%) and muscle pain (21%). Grade 3/4 AEs occurred in 30 patients, most commonly anemia (5%), neutropenia (3%) and nausea (1%). Patients >70 years old: 115 patients experienced at least one AE of any grade, most frequently fluid retention (59%), anemia (58%), neutropenia (56%), nausea (35%) and fatigue (25%). Grade 3/4 AEs occurred in 23 patients, most commonly anemia (10%), neutropenia (6%), diarrhea (3%) and fatigue (3%).	4 *
Kanda, 2012, Japan [85]	Prospective, single center, observational	To clarify the long-term outcomes of imatinib therapy in Japanese patients	70 patients with advanced GIST, 54% male, median age of 64 years (range 39–85)	37 patients received imatinib 400 mg/day, 28 patients 300 mg/day and 5 patients <300mg/day for a median duration of 45 months (max 122)	CTCAE v2.0 Any grade AEs occurring in ≥ 10% and grade 3/4 AEs	69 patients reported at least one AE of any grade, most frequently edema (89%), anemia (81%), hypophosphatemia (73%), leukopenia (69%) and hypocalcemia (66%). Grade 3/4 AEs occurred in 49 patients, including hypophosphatemia (26%), anemia (13%), leukopenia (11%), rash (11%) and neutropenia (9%).	4 *
Li, 2012, China [86]	Prospective, single center	To investigate the efficacy and safety of imatinib dose escalation in Chinese patients	52 patients with advanced GIST, 65% male, mean age of 54 years (±14.0)	Imatinib 600 mg/day, in 14 patients further escalated to 800 mg/day after progression, for a duration of 3.0–56.0 months	CTCAE v2.0 Grade 1/2 and grade 3/4 AEs	600mg All 52 patients on imatinib 600 mg/day had at least one AE. The most common grade 1/2 AEs were edema (81%), fatigue (62%), granulocytopenia (37%), skin rash (23%), nausea (21%) and abdominal pain (21%). The most reported grade 3/4 AEs were granulocytopenia (6%), skin rash (4%) and anemia (4%). All 14 patients on imatinib 800 mg/day had grade 1/2 AEs, mostly edema (100%), fatigue (64%), skin rash (50%), abdominal pain (50%), nausea (43%) and anorexia (43%). Among these patients, 6 had grade 3/4 AEs, mostly fatigue (36%).	4 *
Zhu, 2010, China [87]	Prospective, single center, following a previous publication of Zhu [88]	To further observe the effectiveness of the imatinib treatment on the recurrent GIST and the correlation between the liver metastasis and the outcome	42 patients with recurrent or/and metastatic GIST after the first radical resection, 64% male, median of 52 years (range 23–87)	Imatinib 400 mg/day for an unknown duration	CTCAE v2.0 Grade 1, 2, 3 and 4 AEs	The most common grade 1/2 AEs were edema (59%), nausea (33%), fatigue (33%), anemia (31%), neutropenia (24%) and rash (24%). Two grade 3 AEs were reported, including anemia (2%) and fatigue (2%). There were no grade 4 AEs.	4 *
Xia, 2010, China [89]	Prospective, single center, randomized	To evaluate the effectiveness of resecting liver metastases of GISTs, when performed in conjunction with imatinib treatment	19 patients with GIST and liver metastases, 53% male, median age 53 years (range 31–68)	Neo-adjuvant imatinib 400 mg/day for 6 months prior to resection + adjuvant imatinib 400 mg/day for 2–4 weeks	Unknown AEs of all grade	15 patients experienced AEs, most commonly depigmentation (68%), edema (58%), leucopenia (26%), nausea/vomiting (26%), and diarrhea (21%).	4 *
			20 patients with GIST and liver metastases, 55% male, median age 55 years (range 29–73)	Imatinib 400 mg/day for un unknown duration	Unknown AEs of all grade	17 patients experienced AEs, most frequently depigmentation (75%), edema (50%), diarrhea (30%), nausea/vomiting (30%) and hepatic dysfunction (20%).	
Italiano, 2013, France [90]	Retrospective, international, multicenter	To evaluate the management and outcome of very elderly (age ≥75 years) patients	44 very elderly patients with unresectable and/or metastatic GIST, 52% male, median age of 78 years (range 75–86)	Imatinib 200 mg/day (*n* = 1), 400 mg/day (*n* = 40), 600 mg/dag (*n* = 1) or 800 mg/day (*n* = 2), for an unknown duration	CTCAE v3.0 Grade 1/2 and grade 3/4 AEs	36 patients experienced at least one AE. The most common grade 1/2 AEs were edema (45%), asthenia (43%), nausea/vomiting (25%), diarrhea (20%) and myalgia (18%). Grade 3/4 AEs occurred in 16 patients, most frequently rash (9%), edema (7%) and myalgia (5%).	5 *
Ruka, 2005, Poland [91]	Retrospective, multicenter	To analyze the clinical outcomes of imatinib treatment in inoperable/metastatic GIST in Polish institutions collaborating in the Clinical GIST Registry	165 patients with inoperable/metastatic GIST, 53% male, median age 56 year (range 17–83)	Imatinib 400–800 mg/day for an unknown duration	Unknown Most common and grade 3/4 AEs	The most common AEs were fluid retention, edema, nausea, abdominal and muscle pain, diarrhea and anemia. Reported grade 3/4 AEs were neutropenia (2%), ascites (1%), skin rash (1%) and soft tissue infection (1%).	4 *
Serrano, 2019, Spain [92]	Retrospective, multicenter	To identify clinicopathological and molecular features in long-term responders to imatinib in comparison with patients with GIST reaching the usual median PFS, and to provide further clinical insights from this subgroup collected during the long-term follow-up	64 patients with unresectable or metastatic GIST, 59% male, median age of 62 years	Imatinib in any dose for a median duration of 8 years (range 5.3–10.4)	Unknown Grade 1, 2, 3, and 4 AEs	15 of the 58 long-term responders experienced new emerging AEs after ≥5 years on treatment, most were grade 1/2 including anemia (7%), fatigue (5%), renal failure (5%), diarrhea (3%) and muscle cramps (3%). There were 4 grade 3 AEs including edema (3%), bilateral osteonecrosis of femoral head (2%) and anemia (2%).	5 *
Wong, 2008, UK [93]	Retrospective, multicenter	To assess the effectiveness and toxicity of imatinib and to compare these results with published data	39 patients with advanced unresectable GIST, 64% male, median age of 65 years (range 27–89)	Imatinib 400mg/day and 400 mg twice a day (*n* = 4) at time of progression for an unknown duration	Unknown No grades	21 patients experienced AEs, most frequently peri-orbital edema (23%), nausea/vomiting (13%), skin rash (10%) and diarrhoea (10%).	4 *
Sawaki, 2014, Japan [94]	Retrospective, multicenter	To investigate the effect of imatinib rechallenge on overall survival after failure of imatinib and sunitinib	26 patients with locally advanced or metastatic GIST after failure of imatinib and sunitinib, 62% male, median age of 58 years (range 48–73)	14 patients received imatinib 400mg/day for an unknown duration	CTCAE v3.0 Grade 1, 2 and 3/4 AEs	The most common grade 1/2 AEs were edema (93%), leukopenia (43%), nausea (29%), anorexia (21%), diarrhea (21%) and neutropenia (21%). No grade 3/4 AEs were reported.	4 *
Ogata, 2014, Japan [95]	Retrospective, multicenter	To assess the efficacy of imatinib against advanced or recurrent GIST in Japanese patients	41 patients with unresectable or postoperative recurrent GIST, 66% male, mean age of 63 years (SD ± 13.3)	Imatinib 400 mg/day (*n* = 22) for a median duration of 19.6 months or imatinib 200–300 mg/day (*n* = 19) for a median duration of 41.5 months	Unknown No grades	The most common AEs were edema (59%), fatigue (59%), skin rash (17%), and nausea (12%). No treatment-related deaths occurred.	5 *
Schindler, 2004, Germany [96]	Retrospective, single center	To review patients with GIST who are being treated with imatinib and compare them to a pre-imatinib era group	14 patients with metastatic or locally recurrent GIST, 71% male, median age of 63 years	Imatinib 200mg/day (*n* = 1), 300 mg/day (*n* = 1), 400 mg/day (*n* = 10) or 600mg/day (*n* = 2) for a mean duration of 22.3 months (range 2–39)	Unknown No grades	AEs were reported in 5 patients, including initial nausea (21%), skin rash (7%), preorbital edema (7%), muscular weakness (7%).	4 *
Kasper, 2006, Germany [97]	Retrospective, single center	To assess the response and survival of patients treated with imatinib in palliative and neo-adjuvant clinical setting	16 patients with advanced and overtly metastatic GIST, 63% male, median age was 60 years (range 35–83)	Imatinib 400–800 mg/day in neo-adjuvant (*n* = 3) or palliative (*n* = 13) setting, for an unknown duration	WHO No grades	The most frequent AEs were periorbital edema (38%), skin rash (19%), peripheral edema (19%), alopecia (19%) and diarrhea (19%). No serious adverse events occurred.	4 *
Saito, 2013, Japan [98]	Retrospective, single center	To report the retrospective analysis of recurrent and unresectable GIST patients with imatinib treatment	20 recurrent and unresectable GIST patients, 80% male, median age of 66 years (range 41–86)	Imatinib 400 mg/day (*n* = 14), 300 mg/day (*n* = 4) or 200 mg/day (*n* = 2) for a median duration of 40 months (range 2.5–103)	CTCAE v4.0 Common and grade 3/4 AEs	Frequent reported AEs were fatigue (65%), edema (35%), diarrhea (35%), nausea (25%) and skin rash (10%). The most common grade 3/4 AEs were anemia (15%) and neutropenia (10%).	4 *
Chen, 2005, China [99]	Retrospective, single center	To evaluate the factors determining early recurrence, prognostic factors for giant GISTs, and the effect of imatinib on recurrent GISTs	23 patients with tumor recurrence after surgical resection of giant (>10 cm) GISTs, 48% male, median age of 56 years (±16.9)	9 patients received imatinib 400mg twice a day	Unknown No grades	8 patients reported AEs, most frequently edema (56%) and skin rash (44%).	4 *
Fu, 2018, China [100]	Retrospective, single center	To assess the adverse reactions caused by TKI treatment	36 patients with unresectable or metastatic GIST, 64% male, median age of 56 years (range 36–78)	Imatinib 400 mg/day for a median of 10.5 months (range 1–81)	CTCAE v3.0 Any grade and grade 3/4 AEs	The most common AEs of any grade were skin color change (56%), edema (39%), fatigue (17%), appetite loss (11%), leukopenia (11%) and thrombocytopenia (11%). There were 3 grade 3/4 AEs, including edema (3%), rash (3%) and thrombocytopenia (3%).	4 *
Hsu, 2014, Taiwan [101]	Retrospective, single center	To compare the effectiveness and safety of imatinib dose escalation versus directly switching to sunitinib	63 metastatic GIST patients who had progression on imatinib 400 mg/day, 67% male, median age of 57 years (range 24–83)	Imatinib dose escalation to 600 mg/day (*n* = 21) or 800 mg/day (*n* = 42), for an unknown duration	CTCAE v3.0 All grades and grade 3/4 AEs	The most common AEs of all grades were anemia (64%), leukopenia (27%), neutropenia (27%), edema (24%) and bleeding (22%). The most frequent grade 3/4 AEs were anemia (41%), bleeding (14%), infection (11%) and diarrhea (5%).	4 *
Hsiao, 2006, Taiwan [102]	Retrospective, single center	To report our experience of managing metastatic GIST with imatinib therapy	14 GIST patients with advanced or metastatic disease treated with imatinib, 71% male, median age of 51 years (range 36–72)	Imatinib 400 mg/day (*n* = 13) or 600 mg/day (*n* = 1), for a median duration of 21 months (range 2–44)	CTCAE v? The 4 most common AEs	The most common AEs were edema of the periorbital area and/or legs (65%), abdominal pain (57%), gastrointestinal disturbance (21%) and muscle cramping (?%). There were no grade 3 or 4 AEs.	4 *
Hung, 2019, Vietnam [103]	Retrospective, single center	To assess the efficacy of imatinib	188 patients with unresectable or recurrent GIST, 65% male, median age of 56 years (range 25–84)	Imatinib 400 mg/day for an unknown duration	CTCAE v2.0 Grade 3 and 4 AEs	The most frequent grade 3 AEs were anemia (13%), periorbital edema (8%), diarrhea (7%) and neutropenia (6%). Grade 4 AEs included fatigue (1%), diarrhea (1%), dermatology/skin (15).	4 *
Park, 2009, Korea [104]	Retrospective, single center	To evaluate the efficacy and safety of imatinib dose escalation after disease progression on standard-dose imatinib in Korean patients	24 patients with advanced GIST after disease progression on imatinib 400 mg/day, 75% male, median age of 52 years (range 31–73)	Imatinib 600 mg/day (*n* = 8) or 800 mg/day (*n* = 16) for an unknown duration	CTCAE v3.0 Grade 1, 2, 3 and 4 AEs	The most common grade 1 and 2 AEs were edema (92%), fatigue (83%), anemia (63%), nausea (63%) and alopecia (46%). The most frequent grade 3 AEs were anemia (21%), fatigue (8%) and hyperbilirubinemia (8%). One patient experienced grade 4 anemia (4%).	4 *
Yoo, 2013, Korea [105]	Retrospective, single center	To determine the efficacyand safety of imatinib 800 mg/day as second-line therapy in Asian patients	84 patients with advanced GIST after failure of the standard dose, 63% male, median age of 58 years (range 31–77)	Imatinib 800 mg/day for an unknown duration	CTCAE v3.0 Grade 1/2 and grade 3/4 AEs occurred in ≥10%	All patients experienced AEs. The most common grade 1/2 AEs were anemia (73%), edema (65%), thrombocytopenia (48%), neutropenia (39%) and fatigue (35%). The most frequent grade 3/4 AEs were anemia (26%), neutropenia (11%), hemorrhage (5%), diarrhea (4%) and edema (4%).	5 *
Suresh Babu, 2017, India [106]	Retrospective, single center	To review the clinical data and evaluate the influence of potential prognostic factors on the overall and progression-free survival	44 patients with metastatic GIST, 52% male, median age of 48 years (range 26–67)	Imatinib 400 mg/day, for an unknown duration	CTCAE v4.0 Grade 1/2 AEs	The most common grade 1/2 AEs were facial hyperpigmentation (39%), periorbital edema (36%), muscle cramps (16%) and diarrhea (9%). No patients developed serious AEs.	4 *
Borunda, 2016, Mexico [107]	Retrospective, single center	To report the experience in a highly specialized oncology center in the systemic management of the GIST	71 patients with metastatic, non-resectable or recurrent GIST, 41% male, median age of 58 years	Imatinib 400 mg/day with for a median duration of 2 months (95% CI 0.3–33.0)	CTCAE v? The 3 most common AEs grade 1, 2, 3, 4 and 5	The most common AEs of any grade were fluid retention (83%), fatigue (80%) and diarrhea (60%). There were 3 drug-related deaths including one case due to fluid retention.	4 *
**Imatinib in unknown or combined settings**							
**Author, Year, Country (Ref)**	**Design**	**Aim**	**Patients Characteristics (Number of Patients, Gender Male %, Age Range)**	**Treatment/Intervention**	**Outcome Measure**	**Results**	**Quality Score**
Zhang, 2018, China [108]	Prospective, single center, observational, phase IV	To explore the association of imatinib plasma concentration with adverse drug reactions (ADRs) and influences of genetic polymorphisms on ADRs	129 GIST patients with intermediate or high risk of recurrence, 43% male, mean age of 57 years (range 29–75)	Imatinib 200 mg/day (*n* = 6), 300 mg/day (*n* = 21), 400 mg/day (*n* = 90) or 600 mg/day (*n* = 12) for an unknown duration	CTCAE v3.0 Any grade and grade 3 AEs	The most common AEs of any grade were edema (52%), leukopenia (40%), rash (25%), vomiting (10%) and diarrhea (9%). Edema, vomiting, and fatigue were all significantly correlated with imatinib plasma concentration. Grade 3 AEs were rare, including interstitial pneumonia (2%), anemia (1%), and 2 (2%) patients experienced hepatic dysfunction, rash and edema.	4 *
Xia, 2020, China [109]	Prospective, single center	To evaluate the distribution of imatinib Cmin at different doses and the correlation of adverse reactions with imatinib Cmin	307 patients with GIST treated in adjuvant (*n* = 218) or advanced (*n* = 89) setting, 57% male, median age of 56 (range 23–80)	233 patients received imatinib 400 mg/day and 74 patients received other doses for an unknown duration	CTCAE v4.0 The 10 most common AEs of grade 0, 1, 2 and ≥3	The most common grade 1/2 AEs were periorbital edema (77%), muscle cramps (32%), leukopenia or neutropenia (31%), anemia (26%), and edema of the lower limb (19%). The most frequent grade ≥3 AEs were anemia (2%), leukopenia or neutropenia (2%), rash (1%) and edema of the lower limbs (1%).	4 *
Azribi, 2009, UK [110]	Retrospective, multicenter	To evaluate the efficacy of imatinib in day-to-day clinical setting	36 patients with metastatic, locally advanced inoperable or localized GIST, 47% male, median age 70 years (range 37–86)	Imatinib 400 mg/day, 2 patients started at 200 mg/day, for a median duration of 15.8 months	CTCAE v3.0 Mild and grade 3/4 non-hematological AEs	22 patients experienced clinical AEs. The most common mild AEs were periorbital edema (25%), nausea (11%), diarrhea (8%) and rash (8%). Reported grade 3/4 AEs were nausea/vomiting (8%) and cardiac toxicity (6%).	4 *
Farag, 2017, The Netherlands [111]	Retrospective, multicenter	To assess differences in treatment strategies between elderly patients (aged ≥75 years) and younger patients (<75 years old) with GIST	145 elderly patients, 51% male, median age of 78 years (range 75–92)	85 elderly patients received imatinib 400 mg/day in different treatment settings for an unknow duration	CTCAE v4.0 Grade ≥3 AEs	AEs occurred in 60 patients, of which 55% were grade 1/2 AEs. A total of 22 patients experienced grade ≥3 AEs, most frequently anemia (11%), skin toxicity (6%), infection (4%), fatigue (2%) and dyspnea (2%).	5 *
			665 non-elderly patients, 54% male, median age of 60 years (15–74)	415 non-elderly patients received imatinib 400 mg/day in different treatment settings for an unknow duration	CTCAE v4.0 Grade ≥3 AEs	AEs occurred in 286 patients, of which 59% were grade 1/2 AEs. A total of 84 patients experienced grade ≥3 AEs, most commonly anemia (4%), infection (2%), increased creatinine (2%), nausea (2%) and gastrointestinal hemorrhage (2%).	
Peixoto, 2018, Portugal [112]	Retrospective, single center	To evaluate the evolution in the treatment and prognosis of patients with GISTs since the start of imatinib	131 patients with GIST, 45% male, mean age of 64 years (SD ± 14)	32 patients received imatinib, mostly in adjuvant setting (*n* = 22), in an unknown dose and for an unknown duration	Unknown No grades	24 patients experienced AEs including edema (22%), nausea (9%), fatigue (9%) and diarrhea (9%). Serious AEs were rare, notably hepatotoxicity (3%) and neutropenia (3%).	4 *
Yildrim, 2017, Turkey [113]	Retrospective, single center	To explore the characteristics, prognostic factors and treatment results of GIST cases	35 GIST patients, 49% male, median age of 54 years (range 36–81)	Imatinib 400 mg/day in neo-adjuvant (*n* = 3), adjuvant (*n* = 18) or metastatic (*n* = 4) setting, for an unknown duration	Unknown No grades	No serious AEs occurred, the most common AEs were fatigue (37%), nausea (24%) and edema (24%).	4 *
Ladha, 2008, Pakistan [114]	Retrospective, single center	To assess the response of imatinib in patients with GIST from Pakistan	16 patients of which 12 patients had metastatic disease, 88% male, median age of 52 years (range 38–75)	Imatinib 400 mg/day for un unknow duration	Unknown No grades	10 patients experienced AEs including facial swelling (44%), loose stools (19%), rashes (19%), muscle cramps and body aches (6%).	4 *
Yin, 2016, China [115]	Retrospective, single center	To investigate the efficiency and safety of imatinib in the lower dose in patients with GIST who cannot tolerate imatinib in the standard dose	18 patients in adjuvant (*n* = 12) or advanced/metastatic setting (*n* = 6), 44% male, median age of 52 years (range 34–88)	All 18 patients started at imatinib 400 mg/day, 9 patients continued 400 mg/day for a median duration of 6.2 months (range 1–26) and 9 patients had a dose reduction to 300 mg/day for a median duration of 9.2 months (range 4–17)	CTCAE version unknown Grade 1/2 and grade 3/4 AEs were reported	Reduced dose: Grade 1/2 AEs were edema (33%) and diarrhea (22%). Only one patient experienced grade 3/4 rash (11%). Standard dose: Most frequent reported grade 1/2 AEs were edema (56%), rash (17%), diarrhea (11%), nausea (11%) and liver function abnormality (11%). A total of 4 patients experienced grade 3/4 AEs including liver function abnormality (6%), rash (6%), diarrhea (6%) and nausea (6%).	5 *
Park, 2016, Korea [116]	Retrospective, single center	To evaluate the incidence of imatinib-associated skin rash, the interventional outcomes of severe rash, and impact of severe rash on the outcomes of imatinib treatment in GIST patients	42 (out of 620) patients receiving imatinib developed a severe skin rash, 48% male, median age 63 years (30–81)	Imatinib 300 mg/day (*n* = 2), 400 mg/day (*n* = 38), 600 mg/day (*n* = 1) or 800 mg/day (*n* = 1)	Unknown Grades 3 and 4	Severe skin rash occurred after a median treatment duration of 2.8 months, 40 patients had grade 3 and 2 patients grade 4 skin rash requiring major interventions defined as systemic steroid use (*n* = 17), imatinib dose modification/interruption (*n* = 14) or both (*n* = 9). A total of 28 patients successfully continued imatinib after/with interventions.	4 *
**Sunitinib**							
**Author, Year, Country (Ref)**	**Design**	**Aim**	**Patients Characteristics (Number of Patients, Gender Male %, Age Range)**	**Treatment/Intervention**	**Outcome Measure**	**Results**	**Quality Score**
Demetri, 2006, USA [117]	Prospective, international, multicenter, randomized, phase III	To assess the efficacy and safety of sunitinib	312 patients with advanced GIST after failure of imatinib, 63% male, median age 57 years (range 23–84)	207 patients received sunitinib 50 mg/day in a 4-weeks-on-2-weeks-off schedule for a median duration of 56 days (range 1–236)	CTCAE v3.0 Grade 1/2, 3 and 4 AEs occurring ≥5%	The most common grade 1/2 AEs were anemia (58%), leukopenia (52%), neutropenia (43%), lymphopenia (40%) and thrombocytopenia (36%). The most frequent grade 3 AEs were lymphopenia (9%), neutropenia (8%), fatigue (5%), hand–foot syndrome (4%) and thrombocytopenia (4%). The most common grade 4 AE was neutropenia (2%).	5 *
Demetri, 2012, USA [118]	Prospective, international, multicenter, randomized, phase III	To analyze long-term survival and clinical outcomes of sunitinib	361 advanced GIST patients after imatinib failure, 62% male, median age 56 years (range 23–84)	243 patients received sunitinib 50 mg/day in a 4-weeks-on-2-weeks-off schedule for a median duration of 22 weeks (range 0.4–170)	CTCAE v3.0 Grade 1/2 occurring ≥15% and 3/4 AEs	The most common grade 1/2 AEs were leukopenia (56%), anemia (55%), neutropenia (47%), diarrhea (38%) and thrombocytopenia (37%). The most frequent grade 3/4 AEs were neutropenia (12%), lymphocytopenia (12%), fatigue (10%) and hypertension (8%).	4 *
Reichardt, 2012, Germany [75]	Prospective, international, multicenter, randomized, phase III	To investigate the efficacy of nilotinib versus best supportive care with or without a TKI	248 patients with advanced GIST following prior imatinib and sunitinib failure, 60% male, mean age of 58 years (range 18–83)	23 patients received sunitinib 50 mg/day in a 4-weeks-on-2-weeks-off schedule or 37.5 mg/day continuously for a median duration of 141 days	CTCAE v3.0 Any grade AEs occurring in ≥10% and grade 3/4 AEs occurring in >1%	21 patients experienced at least one AE of any grade, most commonly diarrhea (30%), abdominal pain (26%), headache (17%), fatigue (17%), nausea (13%), vomiting (13%) and rash (13%). Grade 3/4 AEs occurred in 5 patients including neutropenia (9%), thrombocytopenia (4%), diarrhea (4%) and fatigue (4%).	4 *
Adenis, 2014, France [31]	Prospective, multicenter, randomized, phase II	To evaluate the safety and efficacy of masitinib versus sunitinib	44 advanced GIST patients with progressive disease on imatinib ≥ 400 mg/day, 50% male, mean age 64 years (range 31–85)	21 patients received sunitinib 50 mg/day in a 4-weeks-on-2-weeks-off schedule for a median of 3.8 months	CTCAE v4.2 AEs occurring in ≥15%	All patients experienced AEs, most frequently asthenia (67%), diarrhea (57%), rash (57%), edema (43%), nausea/vomiting (33%), abdominal pain (33%), hypertension (33%) and thrombocytopenia (33%).	3 *
George, 2009, USA [119]	Prospective, international, multicenter, randomized, phase II	To assess the antitumor activity, safety, pharmacokinetics and pharmacodynamics of continuous daily sunitinib dosing and to assess morning dosing versus evening dosing	60 patients with imatinib- intolerance or resistant GIST, 47% male, median age of 59 years (range 24–84)	30 patients received morning and 30 patients received evening dosing of sunitinib 37.5 mg/day for a median duration of 46 weeks range (2–93)	CTCAE v3.0 Any grade AEs occurring in ≥15% and grade 1/2, 3, and 4 AEs	Almost all patients (98%) experienced at least one AE of any grade, most commonly anemia (83%), leukopenia (78%), neutropenia (57%), lymphocytopenia (48%), thrombocytopenia (40%) and diarrhea (40%). The most common grade 3 AEs were lymphocytopenia (25%), neutropenia (13%), leukopenia (12%), asthenia (10%), diarrhea (8%) and hypertension (8%). There were 2 cases of grade 4 anemia and 1 case of grade 4 lymphocytopenia.	2 *
Komatsu, 2015, Japan [120]	Prospective, multicenter	To expand the sunitinib safety database in Japanese GIST patients	470 imatinib-resistant or intolerant GIST patients, 63% male, median age of 64 years (range 17–88)	Sunitinib 50 mg/day (*n* = 413), 37.5 mg/day (*n* = 41), 25 mg/day (*n* = 14) in a 4-weeks-on-2-weeks-off schedule, 2 patients received other doses for an unknown duration	CTCAE v3.0 All grades occurring in >10% and grade ≥3 AEs	447 patients experienced at least one AE of all grades, most frequently thrombocytopenia (66%), leukopenia (49%), hand–foot syndrome (45%), hypertension (35%) and neutropenia. Grade ≥ 3 AEs were reported in 329 patients and included thrombocytopenia (33%), neutropenia (22%), leukopenia (15%), anemia (12%) and hypertension (11%).	5 *
Reichardt, 2015, Germany [121]	Prospective, international, multicenter	To provide sunitinib to patients with GIST who were otherwise unable to obtain it and to collect broad safety and efficacy data from a large population	1124 patients with advanced GIST after imatinib failure, 60% male, median age of 59 years (range 10–92)	Sunitinib 50 mg/day in a 4-weeks-on-2-weeks-off schedule or 37.5 mg/day continuously for a median duration of 7.0 months (range <0.1 to 75.4)	CTCAE v3.0 Any grade AEs occurring in ≥10%, grade 1/2, 3 and 4 AEs	1030 of the 1124 patients experienced AEs of any grade, most frequently fatigue (42%), diarrhea (40%), hand–foot syndrome (32%), nausea (29%) and decreased appetite (27%). The most common grade 3 AEs were hand–foot syndrome (11%), fatigue (8%), neutropenia (7%), hypertension (6%) and diarrhea (5%). The most common grade 4 AEs were anemia (2%), thrombocytopenia (1%) and neutropenia (1%). A total of 17 grade 5 AEs were reported.	4 *
Rutkowski, 2012, Poland [122]	Prospective, single center	To analyze the outcomes and factors predicting results of sunitinib therapy	137 patients with inoperable/metastatic GIST after imatinib failure, 54% male, median age of 55 years (range 15–82)	Sunitinib 50 mg/day in a 4-weeks-on-2-weeks-off schedule for an unknown duration	CTCAE v3.0 The 12 most common AEs of any grade and grade 3/4	127 patients experienced at least one AE of any grade, most frequently fatigue (65%), hypertension (43%), hand–foot syndrome (40%), anemia (37%) and neutropenia (36%). Grade 3/4 AEs were reported in 43 patients including fatigue (9%), anemia (6%), neutropenia (5%), diarrhea (3%) and hypertension (3%).	5 *
Rutkowski, 2018, Poland [84]	Prospective, mulicenter, observational	To analyze the treatment results of advanced GIST in the largest, homogenous series of older patients	232 patients with advanced GIST after imatinib failure of which 56 patients were ≥70 years, sex unknown, median age of 55 years (range 15–82)	Sunitinib 50 mg/day in a 4-weeks-on-2-weeks-off schedule for an unknown duration	CTCAE v4.0 Grade 3/4 AEs	Grade 3/4 AEs occurred with similar frequency in both groups. The most common grade 3–4 AEs were fatigue, hypertension, hand–foot syndrome, hypothyroidism, diarrhea, mucositis, anemia and neutropenia.	4 *
Sahu, 2015, India [123]	Prospective, single center	To evaluate the efficacy and safety of sunitinib	15 patients with imatinib-resistant locally advanced or metastatic GIST, 67% male, median age of 48 years (range 26–69)	Sunitinib 50 mg/day in a 4-weeks-on-2-weeks-off schedule for a median duration of 10 cycles (range 1–47)	CTCAE v3.0 Grade 3 and 4 AEs	6 patients experienced grade 3 AEs, most frequently hand–foot syndrome (20%), hypertension (13%), anemia (13%) and thrombocytopenia (13%). No grade 4 AEs were reported.	4 *
Shen, 2017, China [124]	Prospective, multicenter, phase IV	To determine the efficacy and safety of sunitinib in Chinese patients	59 patients with imatinib resistant or intolerant GIST, 66% male, mean age of 55 years (range 29–82)	Sunitinib 50 mg/day in a 4-weeks-on-2-weeks-off schedule for a median duration of 223 days (range 28–1288)	CTCAE v3.0 All grade AEs occurring in ≥20% and grade 3, 4 and 5 AEs	58 of the 59 patients reported at least one AE. The most common AEs of all grades were leukopenia (64%), fatigue (53%), hand–foot syndrome (51%), neutropenia (49%) and an increased AST (36%). The most frequently reported grade 3 AEs were neutropenia (14%), leukopenia (14%), hand–foot syndrome (10%) and thrombocytopenia (7%). The most common grade 4 AE was neutropenia (5%) and grade 5 AEs were reported in 5 patients.	4 *
Shirao, 2010, Japan [125]	Prospective, multicenter, phase I/II	To evaluate the efficacy, safety, pharmacokinetics, and pharmacodynamics of sunitinib in Japanese patients	30 patients with imatinib resistant or intolerant GIST, 63% male, median age 56 years (range 41–74)	Sunitinib 50 mg/day in a 4-weeks-on-2-weeks-off schedule for a median duration of 4 cycles (range 2–10)	CTCAE v2.0 Any grade AEs occurring in ≥25% and grade 1/2, 3 AEs	The most frequently reported AEs of any grade were neutropenia (90%), thrombocytopenia (90%), hand–foot syndrome (87%), leukopenia (87%) and increased AST (73%). The most common grade 3 AEs were neutropenia (37%), anemia (33%), hand–foot syndrome (30%), lymphocytopenia (30%) and hypertension (23%).	4 *
Desai, 2006, USA [126]	Prospective, single center, observational	To describe the prevalence and clinical presentation of thyroid dysfunction related to sunitinib therapy	42 patients with imatinib-resistant GIST, sex and age were not reported	Sunitinib 50 mg/day in a 4-weeks-on-2-weeks-off schedule for a median duration of 37 weeks (range 10–167)	Unknown	15 patients developed persistent primary hypothyroidism.	4 *
Mannavola, 2007, Italy [127]	Prospective, multicenter, phase 1/2	To evaluate the effect of sunitinib on thyroid function	24 patients with GIST, 54% male, median age unknown (range 40–75)	Sunitinib 50 mg/day in a 4-weeks-on-2-weeks-off schedule for an unknow duration	Unknown	After a median of 3 cycles (range 1–6) 10 of the 24 patients developed hypothyroidism.	4 *
Wolter, 2008, Belgium [128]	Prospective, single center, observational	To define the incidence and severity of hypothyroidism	17 patients with imatinib-refractory or -intolerant GIST, 76% male, median age 61 years (42–74)	Sunitinib 50 mg/day in a 4-weeks-on-2-weeks-off schedule for a median duration of 33 weeks (range 10–82)	Unknown	2 (12%) patients developed a (sub)clinical hypothyroidism requiring treatment.	4 *
Matsumoto, 2011, Japan [129]	Retrospective, single center	To assess the efficacy and safety of sunitinib in Japanese patients	18 patients with advanced GIST who were resistant or intolerant to imatinib, 72% male, median age of 58 years (range 26–77)	Sunitinib 50 mg/day in a 4-weeks-on-2-weeks-off schedule, for a median duration of 3.5 cycles (range 1–14)	CTCAE v2.0 Grade 1/2 AEs occurring in ≥5%, grade 3 and 4 AEs	The most frequent grade 1/2 AEs were hand–foot syndrome (89%), liver dysfunction (72%), fatigue (56%), neutropenia (56%) and anemia (56%). The most common grade 3 AEs were thrombocytopenia (22%), liver dysfunction (17%), fatigue (11%), neutropenia (11%) and anemia (11%). The only reported grade 4 AE was liver dysfunction (6%).	4 *
Den Hollander, 2019, The Netherlands [130]	Retrospective, international, multicenter	To investigate predictive factors for grade 3 or 4 sunitinib-related toxicities and for PFS and OS in a population treated outside a clinical trial	91 patients with irresectable or metastatic GIST who had progression or intolerance on imatinib, 65% male, median age of 59 years (range 19–85)	16 patients received sunitinib 50 mg/day in a 4-weeks-on-2-weeks-off schedule, 65 patients received 37.5 mg/day continuously and 10 patients received a lower dosing schedule, for a median duration of 9.3 months (range 0.3–84.2)	CTCAE v4.0 Grade 3 and 4 AEs	Grade 3 or 4 AEs were observed in 51 patients, most commonly grade 3 diarrhea (13%), grade 3 neutropenia (12%), grade 3 asthenia, (10%), grade 3 hypertension (10%) and grade 3 hand–foot syndrome (8.8%). There were no treatment-related deaths.	5 *
Kefeli, 2013, Turkey [131]	Retrospective, multicenter	To evaluate the efficacy and tolerability of sunitinib therapy in Turkish patients	57 GIST patients who had progressive disease or experienced unacceptable toxicity during imatinib, 58% male, median age of 55 years (range 16–84)	Sunitinib 25–50 mg/day, for an unknown duration	Unknown Any grade and grade 3 AEs	AEs were reported in 78% of the patients. The most common AEs of any grade were anemia (48%) and fatigue (31%). The most common grade 3 AE was hand–foot syndrome (19%). Hypothyroidism occurred in only one patient.	5 *
Lee, 2009, Korea [132]	Retrospective, single center	To obtain an accurate description of cutaneous effects observed with sunitinib use	119 patients receiving sunitinib for treatment of renal cell carcinoma (*n* = 79) or GIST (*n* = 40), 63% male, mean age of 59 years (range 30–87)	Sunitinib 50 mg/day in a 4-weeks-on-2-weeks-off schedule, for an unknown duration but at least 3 months	CTCAE v3.0 Only cutaneous AEs	Most frequent cutaneous AEs in GIST patients were hand and-foot skin reaction (43%), stomatitis (43%), facial swelling (18%), yellowish facial discoloration (18%), hair depigmentation (10%) and erythematous eruption on the trunk (10%).	4 *
Yoon, 2012, Korea [133]	Retrospective, single center	To assess the efficacy and safety of sunitinib with regards to primary genotypes of tumor in Korean patients with advanced GISTs	88 patients with advanced GISTs who failed initial therapy of imatinib, 63% male, median age of 59 years (range 25–76)	74 patients received sunitinib 50 mg/day in a 4-weeks-on-2-weeks-off schedule and 14 patients received sunitinib 37.5 mg/day continuously, for an unknown duration	Unknown Any grade, grade 1/2 and 3/4 AEs	The most frequent AEs of any grade were anemia (89%), thrombocytopenia (73%), neutropenia (69%), elevated bilirubin (64%) and increased AST (61%). The most common grade 3/4 AEs were neutropenia (34%), anemia (33%), thrombocytopenia (33%), hand–foot skin reaction (25%) and decreased albumin level (16%).	4 *
Li, 2012, China [134]	Retrospective, single center	To evaluate the efficacy and safety of sunitinib in Chinese patients	55 patients with advanced GIST who were resistant or intolerant to prior imatinib treatment, 73% male, median age of 54 years (95% CI: 49.8–58.2)	35 patients received sunitinib 50 mg/day in a 4-weeks-on-2-weeks-off schedule for an unknown duration	CTCAE v3.0 Grade 1, 2 and 3/4 AEs	Fractioned dose group: The most common grade 1 and 2 AEs were hand–foot syndrome (64%), fatigue (58%), hypertension (53%), anemia (53%) and neutropenia (50%). The most common grade 3/4 AEs were neutropenia (19%), fatigue (8%), anemia (8%) and thrombocytopenia (8%).	4 *
				19 patients received sunitinib 37.5 mg/day continuously for an unknown duration	CTCAE v3.0 Grade 1, 2 and 3/4 AEs	Continuous dose group: The most common grade 1 and 2 AEs were fatigue (58%), neutropenia (58%), hand–foot syndrome (53%), hypertension (53%) and anemia (47%). The most common grade 3/4 AEs were neutropenia (11%), fatigue (5%), anemia (5%) and thrombocytopenia (5%).	
Fu, 2018, China [100]	Retrospective, single center	To assess adverse reactions caused by TKI treatment	22 patients with unresectable GIST after imatinib treatment failure or intolerable recurrent metastases, 50% male, median age of 52 years (range 36–74)	Sunitinib 50 mg/day in a 4-weeks-on-2-weeks-off schedule or 37.5 mg/day continuously for a median of 12.8 months (range 2–24)	CTCAE v3.0 Any grade and grade 3/4 AEs	The most common AEs of any grade were skin color change (91%), leukopenia (64%), hand–foot skin reactions (55%), fatigue (41%), thrombocytopenia (32%) and hair pigmentation (32%). The most frequent grade 3/4 AEs were hand–foot skin reaction (14%) and leukopenia (9%).	4 *
Chen, 2014, Taiwan [135]	Retrospective, single center	To clarify the efficacy and safety of fractioned dose regimen of sunitinib by a pharmacokinetic and efficacy study	55 patients with recurrent or metastatic GIST who failed prior imatinib therapy, 58% male, median age 55 years (range 15–88)	29 patients received sunitinib 50 mg/day in a 4-weeks-on-2-weeks-off schedule for a median duration of 9.2 months	CTCAE v2.0 All grades and grade 3/4 AEs	Devided dose group: The most common AEs of all grades were anemia (59%), leukopenia (59%), hypertension (59%), thrombocytopenia (55%) and diarrhea (52%). The most frequent grade 3/4 AEs were anemia (31%), GI bleeding (17%), hand–foot syndrome (10%), neutropenia (10%) and thrombocytopenia (10%).	4 *
				26 patients received sunitinib 37.5 mg/day continuously for a median duration of 9.2 months	CTCAE v2.0 All grades and grade 3/4 AEs	Non-divided dose group: The most common AEs of all grades were hand–foot skin reaction (65%), anemia (62%), leukopenia (58%), thrombocytopenia (58%) and hypertension (54%). The most frequent grade 3/4 AEs were hand–foot syndrome (35%), anemia (19%), leukopenia (12%), neutropenia (12%) and hypertension (8%).	
Hsu, 2014, Taiwan [101]	Retrospective, single center	To compare the effectiveness and safety of imatinib dose escalation versus directly switching to sunitinib	28 metastatic GIST patients who had progression or intolerance on imatinib 400 mg/day, 54% male, median age of 59 years (15–91)	Sunitinib in an unknown dose and for un unknown duration	CTCAE v3.0 All grades and grade 3/4 AEs	The most common AEs of all grades were anemia (68%), leukopenia (61%), neutropenia (57%), thrombocytopenia (57%), hypertension (50%), hand–foot syndrome (50%) and diarrhea (50%). The most frequent grade 3/4 AEs were anemia (27%), hand–foot syndrome (25%) and thrombocytopenia (14%).	4 *
Chu, 2007, USA [136]	Retrospective, single center	To determine the cardiovascular risk associated with sunitinib	75 patients with imatinib-resistant metastatic GIST, 68% male, mean age 54 years (±11.5)	All patients received sunitinib, 36 patients received sunitinib 50 mg/day in a 4-weeks-on-2-weeks-off schedule for a median of 33.6 weeks (range 3.3–112.4)	CTCAE v3.0	35 (47%) patients developed hypertension, 8 patients suffered a cardiovascular event with congestive heart failure occurring in 6 (8%) patients, LVEF decline of ≥10% occurred in 10 of the 36 patients treated with the FDA approved dose.	5 *
**Regorafenib**							
**Author, Year, Country (Ref)**	**Design**	**Aim**	**Patients Characteristics (Number of Patients, Gender Male %, Age Range)**	**Treatment/Intervention**	**Outcome Measure**	**Results**	**Quality Score**
Demetri, 2013, USA [13]	Prospective, international, multicenter, randomized, phase III	To evaluate the efficacy and safety of regorafenib	199 patients with metastatic and/or unresectable GIST progressing after failure of imatinib and sunitinib, 64% male, median age of 60 years (range 18–87)	133 patients received regorafenib 160 mg/day for the first 3 weeks of every 4-week cycle for a median duration of 22.9 weeks	CTCAE v4.0 Any grade AEs occurring in ≥10%, grade 3 and 4 AEs	130 of the 132 patients reported at least one AEs of any grade, most frequently hand–foot skin reaction (56%), hypertension (49%), diarrhea (40%), fatigue (39%) and oral mucositis (38%). The most frequent grade 3 AEs were hypertension (23%), hand–foot skin reaction (20%), diarrhea (5%), fatigue (2%) and rash (2%). There were 2 grade 4 AEs, including 1 grade 4 hypertension. Grade 5 AEs were reported in 2 patients.	5 *
Komatsu, 2015, Japan [137]	Prospective, multicenter, randomized, phase III	To assess the efficacy and safety of regorafenib in Japanese patients enrolled in the GRID study	17 patients, 76% male, median age of 54 years (range 27–67)	12 patients received regorafenib 160 mg/day for the first 3 weeks of every 4-week cycle for a median duration of 23 weeks (range 5.7–42.9)	CTCAE v4.0 Any grade AEs occurring in ≥20% and grade ≥3 AEs	All patients receiving regorafenib experienced AEs, the most common AEs of any grade were hand–foot skin reaction (92%), oral mucositis (58%), alopecia (50%), diarrhea (50%), hoarseness (50%), hypertension (50%), rash (50%) and proteinuria (50%). The most common ≥3 AEs were hypertension (25%), hand–foot skin reaction (17%) and rash (17%).	5 *
Ben-Ami, 2016, USA [138]	Prospective, multicenter, phase II, following the publication of George [139]	To assess long-term safety and efficacy of regorafenib	33 patients with metastatic and/or unresectable GIST after failure of at least imatinib and sunitinib, 58% male, median age 56 years (25–76)	Regorafenib 160 mg/day for the first 3 weeks of every 4-week cycle for a median duration of 15 cycles (range 1–45 cycles)	CTCAE v4.0 Any grade AEs occurring ≥25% and grade 1, 2, 3 and 4 AEs	All 33 patients experienced at least one AE of any grade, most commonly hand–foot skin reaction (91%), fatigue (85%), diarrhea (79%) and hypertension (76%). The most frequent grade 3 AEs were hypertension (39%), hand–foot skin reaction (36%), hypophosphatemia (18%), rash (12%), diarrhea (9%) and abdominal pain (9%). The most frequent grade 4 AE was hyperuricemia (6%).	4 *
Son, 2017, Korea [140]	Prospective, multicenter	To confirm the efficacy and safety of regorafenib for advanced GISTs reported in the GRID phase III trial in Korean patients	56 patients with advanced GIST, 60% male, median age 56 years (range 50–62)	Regorafenib 160 mg/day for the first 3 weeks of every 4-week cycle for a median duration of 5 cycles (range 1–29)	CTCAE v4.0 Any grade AEs occurring in ≥5% and grade 3 and 4 AEs	55 of the 56 patients experienced at least one AE of any grade, most frequently hand–foot skin reaction (82%), fatigue (54%), oral mucositis (44%), alopecia (35%) and hoarseness (33%). The most common grade 3 AEs were hand–foot skin reaction (25%), hypertension (7%), skin rash (7%) and fatigue (4%). There were no grade 4 AEs or treatment-related deaths.	4 *
Kim, 2019, Korea [141]	Prospective, single center, phase II	To assess the efficacy and safety of a continuous daily dosing schedule of regorafenib	25 patients with GIST after failure of imatinib and sunitinib, 84% male, median age of 60 years (range 42–74)	Regorafenib 100 mg/day continuously in a 4-week cycle for a median duration of 6 cycles (range 2–16)	CTCAE v4.03 Any grade AEs occurring in ≥10%, grade 3 and 4 AEs	All patients experienced at least one AE of any grade, most commonly hand–foot skin reaction (88%), hoarseness (72%), myalgia (60%), ALT elevation (56%) and diarrhea (48%). The most common grade 3 AEs were hand–foot skin reaction (16%) and ALT elevation (8%). There were no reports of grade 4 AEs.	4 *
Hu, 2020, Taiwan [142]	Prospective, single center, following the publication of Yeh [143]	To assess the efficacy, prognosis and safety of regorafenib in inducing an objective response or stable disease	28 patients with advanced inoperable/metastatic GIST after failure of imatinib and sunitinib, 71% male, median age 61 years (range 36–71)	Regorafenib 160 mg/day for the first 3 weeks of every 4-week cycle for a median duration of 5.5 months	CTCAE v4.0 Any grade, grade 1/2, and 3 AEs	All patients experienced at least one AE of any grade, most commonly hypertension (93%), hand–foot skin reaction (86%), anemia (79%), hepatic toxicity (54%) and thrombocytopenia (32%). The most frequent reported grade 3 AEs were hypertension (21%), hand–foot skin reaction (21%), hepatic toxicity (18%) and anemia (11%).	5*
Kollar, 2014, UK [144]	Prospective, single center, cohort	To assess the safety and efficacy of regorafenib in a routine clinical setting	20 advanced GIST patients who had no other approved therapeutic options, 65% male, median age of 68 years (range 45–87)	Regorafenib 160 mg/day (*n* = 15), 120 mg /day (*n* = 3) or 80 mg (*n* = 2) for the first 3 weeks of every 4-week cycle for a median duration of 9.25 months (range 0.1–15.3)	CTCAE v4.0 Any grade and grade 3/4 AEs	The most common AEs of any grade were fatigue (80%), hand–foot-syndrome (55%), hypertension (50%), diarrhea (50%), oral mucositis (40%) and hoarseness (40%). Grade 3/4 AEs were documented in 50% of the patients including hand–foot-syndrome (15%), hypertension (15%) and skin rash (10%). There were no treatment r-lated deaths.	4 *
Chamberlain, 2020, UK [145]	Prospective, single center, cohort	To evaluate regorafenib toxicities and their management in a real-world GIST population	50 patients with GIST pre-treated with at least two lines of treatment, 64% male, median age of 56 years (IQR 46–66.5)	Regorafenib 160 mg/day (*n* = 42), 120 mg/day (*n* = 3) or 80 mg/day (*n* = 5) for the first 3 weeks of every 4-week cycle for a median duration of 7.6 months (IQR 3.1–12.9)	CTCAE v4.0 Grade 3/4 AEs	Grade 3/4 AEs were seen in 23 patients, including palmar-plantar erythrodysesthesia (18%), fatigue (14%) and hypertension (8%).	4 *
Schvartsman, 2017, USA [146]	Retrospective, single center	To summarize our experience regarding prescribing patterns, efficacy and toxicity of regorafenib	28 GIST patients who had previously progressed on imatinib and sunitinib, 61% male, median age of 58 years (range 21–84)	22 patients received regorafenib 120 mg/day continuously and 6 patients received regorafenib 160mg/day for the first 3 weeks of every 4-week cycle, for a median duration of 7.3 months (range 0.9–18.8)	CTCAE v4.0 AEs of any grade and grade 3/4 AEs	26 patients experienced AEs of any grade, including hand–foot skin reaction (61%), fatigue (50%), weight loss (43%), diarrhea (39%), nausea (25%) and hypertension (25%). Grade 3/4 AEs occurred in 12 patients, most frequently hand–foot skin reaction (18%), fatigue (18%) and weight loss (14%).	4 *
Ivanyi, 2020, Germany [147]	Retrospective, multicenter, cohort	To investigate the incidence and clinical course of regorafenib associated hepatic toxicity in patients with GIST in a real-world setting	21 patients with metastatic GIST, 76% male, median age of 67 years (range 31–87)	Regorafenib in an unknown dose for a median duration of 5.15 months (range 2-20)	CTCAE v4.0 Hepatic-toxicity-related AEs	Hepatic toxicity was identified in 5 patients (23.5%); 4 patients developed laboratory hepatic toxicity and 1 patient only had clinical signs of hepatic toxicity. A total of 1 of the 5 patients exhibited liver progression of GIST at time of hepatic toxicity.	4 *
**Ripretinib**							
**Author, Year, Country (Ref)**	**Design**	**Aim**	**Patients Characteristics (Number of Patients, Gender Male %, Age Range)**	**Treatment/Intervention**	**Outcome Measure**	**Results**	**Quality Score**
Blay, 2020, France [33]	Prospective, international, multicenter, randomized, phase III	To evaluate the safety and efficacy of ripretinib as fourth-line therapy	129 advanced GIST patients with progression on at least imatinib, sunitinib, and regorafenib or intolerance to any of these therapies, 57% male, median age 61 years (range 29–83)	85 patients received ripretinib 150 mg/day in 28-day cycles for an unknown duration	CTCAE v4.03 Grade 1/2, 3, 4 and 5 AEs were reported	The most common grade 1/2 AEs were alopecia (49%), myalgia (28%), nausea (26%), fatigue (22%) and hand–foot syndrome (21%). The most common grade 3 AEs were lipase increase (5%), hypertension (4%), fatigue (2%) and hypophosphataemia (2%). Grade 4 anemia occurred in 1 patient.	5 *

* Represents the overall methodological quality of the study ranging from 1 * to 5 *; 1 * indicating a study of poor quality and 5 * indicating a study of good quality.

## Data Availability

Not applicable.

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
