# Peer review of "Health-Related Quality of Life and Side Effects in Gastrointestinal Stromal Tumor (GIST) Patients Treated with Tyrosine Kinase Inhibitors: A Systematic Review of the Literature"

_cancers, 2022, doi:10.3390/cancers14071832_

Round 1

Reviewer 1 Report

This manuscript entitled “Health Related Quality of Life and Side Effects in Gastro-Intes- tinal Stromal Tumour (GIST) Patients Treated with Tyrosine Kinase Inhibitors: a Systematic Review of the Literature.” by Deborah van de Wal et al. comprehensive reviewed the previous studies of TKIs in GIST in terms of QoL and AEs. The manuscript is well-written and can provide useful information to readers. I have some comments for this manuscript.

  1. Gastrointestinal stromal tumour is more commonly used than “Gastro-Intestinal Stromal Tumour”
  2. Authors may cite some reference for molecular profiling of GIST. (10.3390/cancers11050679)
  3. “Historically, the median survival of patients with metastatic GIST was only a year.” (Line 64) The authors should state when the median survivals…..
  4. Notably, the FDA approvals of imatinib…. (Line 84) It is not surprising that FDA approve the TKIs based on efficacy with acceptable toxicity profile.
  5. In this study, authors mixed the adjuvant, neoadjuvant and metastatic treatment.
  6. Papers selected by either ? either one or both ? (Line 121)
  7. The incidence of HFSR varied significantly per tumor type. (Line 369). Highest incidence in GIST possibly resulted from longest duration of TKI use.
  8. A meta-analysis reported a significant………….. (Lines 373-376) is unclear.

Author Response

Reviewer 1

This manuscript entitled “Health Related Quality of Life and Side Effects in Gastro-Intes- tinal Stromal Tumour (GIST) Patients Treated with Tyrosine Kinase Inhibitors: a Systematic Review of the Literature.” by Deborah van de Wal et al. comprehensive reviewed the previous studies of TKIs in GIST in terms of QoL and AEs. The manuscript is well-written and can provide useful information to readers. I have some comments for this manuscript.

  1. Gastrointestinal stromal tumour is more commonly used than “Gastro-Intestinal Stromal Tumour”
  • We have changed gastro-intestinal stromal tumour to gastrointestinal stromal tumour throughout our manuscript according to the reviewers’ suggestion.
  1. Authors may cite some reference for molecular profiling of GIST. (10.3390/cancers11050679)
  • We would like to thank the reviewer for this comments and have added a sentence on molecular profiling of GIST on page 2 (line 60-62) of our manuscript.

  1. “Historically, the median survival of patients with metastatic GIST was only a year.” (Line 64) The authors should state when the median survivals…..
  • We have changed “historically” in “before the introduction of imatinib” on page 2 (line 65) of our manuscript.

  1. Notably, the FDA approvals of imatinib…. (Line 84) It is not surprising that FDA approve the TKIs based on efficacy with acceptable toxicity profile.
  • We rephrased this sentence to “Nevertheless, until recently, FDA approvals, also for imatinib both in metastatic and adjuvant setting, sunitinib and regorafenib, were only based on objective or physician-reported data.” (Page 2, line 86-88)
  1. In this study, authors mixed the adjuvant, neoadjuvant and metastatic treatment.
  • We did mix different treatment setting in case of imatinib, because imatinib is registered for neo-adjuvant, adjuvant and metastatic treatment of GIST. In the review we wanted to focus on HRQoL impact of different types of TKIs, in which the treatment setting is less relevant. However, in the tables and text, we always state the treatment setting, if this was known.

  1. Papers selected by either ? either one or both ? (Line 121)
  • We would like to thank the reviewer for this remark and we have adjusted “either” to “both reviewers” on page 3 (line 125-126) of the manuscript.

  1. The incidence of HFSR varied significantly per tumor type. (Line 369). Highest incidence in GIST possibly resulted from longest duration of TKI use.
  • There is no evidence for a relationship between incidence of HFSR and previous TKI use or duration of TKI use. The exact molecular mechanisms behind the increased incidence are poorly understood, it is hypothesized that this may be due to the extended spectrum of target kinases (VEGF-R1/-R2/-R3, FGFR-1, PDGFR-α/β, KIT, RET, RAF and p38 MAPK) including TIE-2, that regorafenib inhibits more potently compared to other TKIs. The occurrence of HFSR seems rather dose related.
  • We would like to thank the reviewer for this remark and we added, “There is no evidence for a relationship between incidence of HFSR and previous TKI use or duration of TKI use, the exact molecular mechanisms behind the increased incidence are poorly understood, and the occurrence of HFSR seems rather dose related.” to the manuscript. (Page 27, line 391-394)

  1. A meta-analysis reported a significant………….. (Lines 373-376) is unclear.
  • We have rephrased this sentence to make clear that this meta-analysis focused on regorafenib-associated AEs. (Page 27, line 397-398)

Reviewer 2 Report

Oncologists often wonder which TKI preparation should be administered which is better and has fewer complications. Analyzes by researchers on the side effects of individual drugs are valuable. Researchers have analyzed many studies, most of them concerned with the advers events of the drugs. Many times patients were given different medications depending on the stage of treatment, so it is difficult to judge which medication developed side effects
In the chapter: Regorafenib
"Ten studies investigated...- you should write the number of studies.
I could not find the references number 160 and 164

It is the interesting and valuable manuscript. This manuscript willbe helpful in selecting the appropriate medication for pediatric patients with rare cancer such as GIST.

Author Response

1. In the table above the chapter regorafenib, the 10 studies on regorafenib are described. Given the length of our paper, we did not mention and cite all studies in the text, only the ones we highlighted.

2. We apologize for this mistake. Reference 160 and 164, are now 162 and 166. Both references are cited in the discussion and you can find them on pubmed:

162: https://pubmed.ncbi.nlm.nih.gov/27260018/  

166: https://pubmed.ncbi.nlm.nih.gov/33165751/

Reviewer 3 Report

Interesting review, very well thought out and described. Perhaps it should be done more emphasis in the scarce importance that often the oncologists we do to the side effects of treatments and their implication in quality of life of our patients.

In the text I have not found the meaning of the acronym PROM.

Congratulations for this work.

Author Response

We would like to thank the reviewer for the compliments. We have now explained the meaning of PROM on page 4, line 156-157.

Reviewer 4 Report

In this study, a systematic review of the available literature on HRQoL and side effects of different tyrosine kinase inhibitors (TKIs) registered for the treatment of gastro-intestinal stromal tumours (GIST) was performed.
Analysis of the results of 104 papers on adverse effects and patients HRQoL provided important insights into the relationship between methods of HRQoL assessment, patient-reported adverse effects and treatment regimen. 
The authors of the study suggest that reported side effects have been underestimated by physicians, or that the measures used to assess HRQoL do not capture all of the important issues that determine the HRQoL of a patient with GIST.
This is a very important finding that draws attention to the need to revise the measures used to assess HRQoL for the sake of patient welfare.

Author Response

We would like to thank the reviewer for the compliments.